



# Intrahalocline eddies in the Amundsen Basin observed in the distributed network from the MOSAiC expedition

Alejandra Quintanilla-Zurita[1], Benjamin Rabe[1], Claudia Wekerle[1], Torsten Kanzow[1], Ivan Kuznetsov[1], Sinhue Torres-Valdes[1], Enric Pallàs-Sanz[2], and Ying-Chih Fang[3]

[1]Alfred-Wegener-Institut Helmholtz-Zentrum für Polar- und Meeresforschung, Bremerhaven, Germany
[2]Centro de Investigación Científica y de Educación Superior de Ensenada, Baja California, Mexico
[3]Department of Oceanography, College of Marine Sciences, National Sun Yat-sen University, Kaohsiung, Taiwan

**Correspondence:** Alejandra Quintanilla-Zurita (alejandraquintanillaz@gmail.com)

**Abstract.** Hydrographic and velocity observations from the Multidisciplinary Drifting Observatory for the Study of Arctic Climate (MOSAiC) expedition (2019–2020) reveal the presence of nine intrahalocline eddies (IHEs) in the Amundsen Basin during the winter drift of the Distributed Network (DN). Despite their relevance for Arctic stratification and mixing, IHEs in the Amundsen Basin remain poorly documented. Our study addresses this gap by providing the first detailed characterisation

based on coordinated in situ hydrographic and velocity observations during wintertime. Eddies were identified as isopycnal displacements in Ice-Tethered Profiler (ITP) data. Additionally, by assessing rotational velocity signatures from Acoustic Doppler Current Profiler (ADCP) measurements, we applied a centre-detection method based on maximum swirl velocity (MSV). Nine anticyclonic eddies were observed, with radii ranging from 3.7 to 8.4 km and vertical extents between 23 and 80 m. Most eddies exhibited solid-body rotation in their cores, with maximum azimuthal velocities of up to 0.28 ms$^{-1}$ and localised shal-

lowing of the mixed layer by over 10 m. Water mass analysis showed that the eddy cores contained Eurasian halocline waters with consistent anomalies in temperature, salinity, and density relative to surrounding profiles, allowing us to infer pre-existing stratification conditions and offering clues to their origin. The observed eddy scales lie close to or slightly below the first baroclinic Rossby deformation radius $L_1 \approx 6.9$ km, placing them in the (sub)mesoscale dynamical regime, consistent with quasi-geostrophic behaviour. The MSV method yields systematically larger eddy radius estimates up to 25% greater than tra-

ditional detection techniques that rely on velocity profiles or isopycnal displacements alone. This correction to the radius is essential, as it provides a more realistic measure of eddy size and dynamics under ice-covered conditions and could improve comparability across under-ice eddy studies. Although specific generation mechanisms remain uncertain, thermohaline signatures suggest that local convection and baroclinic instability play a role in their formation. Our results provide new insights into the dynamics of under-ice eddies and their potential impact on Arctic oceanography and climate processes, addressing

essential gaps in understanding polar mesoscale dynamics.





# 1 Introduction

The global ocean surface is densely populated by mesoscale eddies, which can be tracked through satellite-derived sea surface height anomalies (Chelton et al., 2011). However, much less is known about the subsurface eddies below the mixed layer in sea ice-covered regions, in particular, in the Arctic Ocean. Intrahalocline eddies (IHEs)—analogous in some respects to intrathermocline eddies (ITEs, Dugan et al., 1982)—are coherent features, ranging from submesoscale to mesoscale, that are embedded within the halocline, generally just below the mixed layer (e.g., Kuzmina et al., 2008). Unlike their open-ocean counterparts, IHEs are uniquely shaped by Arctic stratification, with potential impacts on halocline stability and cross-basin transport. These subsurface features can modify the mixed layer depth (MLD) and transport anomalous water masses and nutrients across ocean basins (Timmermans et al., 2008; Zhao et al., 2014).

The Arctic Ocean exhibits some of the smallest dynamic scales globally, with the first baroclinic Rossby radius of deformation typically around 10 km (Nurser and Bacon, 2014). However, finer-scale analyses suggest that this limit may be even smaller, making it particularly challenging to sample mesoscale structures beneath the sea ice. This fine scale complicates the detection of IHEs, which may exhibit diameters close to or below this threshold. Therefore, we refer to these features as (sub)mesoscale eddies, to acknowledge that their scales may span both the mesoscale and submesoscale regimes—especially in the Arctic, where overlapping dynamical processes make precise scale separation difficult to define (Della Penna and Gaube, 2019).

The Amundsen Basin is the deepest part of the Arctic Ocean, reaching depths of 4500 m, bounded by the Lomonosov and Gakkel ridges. The Amundsen Basin is characterized by strong stratification in the upper layers, with a mixed layer that extends to a depth of 50 m in winter, temperatures close to the freezing point ($\approx$ -1.8 °C) and salinity <33. Underlain by a sharp halocline that separates the mixed-layer from the warmer and saltier Atlantic water located at $\approx$200 m depth (Rudels et al., 1996; Polyakov et al., 2020). The Transpolar Drift, the primary surface current in the central Arctic Ocean, influences the Amundsen Basin by transporting sea ice and freshwater from the Siberian shelves to the Fram Strait, shaping the large-scale structure of the halocline (Morison et al., 2012; Rabe et al., 2014).

In general, eddies are characterised by vertical displacement of isopycnals that is maximal at their centre, consistent with geostrophic balance, where horizontal velocities are minimal at the eddy core (Zhao and Timmermans, 2015). IHEs are (sub)surface vortices with distinct thermohaline properties relative to ambient waters (Kostianoy and Belkin, 1989). Anticyclonic IHEs tend to have domed isopycnals above and depressed isopycnals below, whereas cyclonic IHEs display the inverse structure—depressed isopycnals in the upper part and domed isopycnals below (McGillicuddy Jr, 2015). Their velocity field is characterised by a subsurface maximum of azimuthal velocity, presenting an azimuthal symmetry (Thomas, 2008). Previous studies have investigated these types of eddies (e.g., Aagaard and Carmack, 1989; Manley and Hunkins, 1985; Timmermans et al., 2008; Zhao et al., 2014), with most of the work focusing on Canada Basin eddies. In contrast, only a few studies have documented Eurasian Basin eddies based on mooring observations (Polyakov et al., 2012; Woodgate et al., 2001). Notably, Zhao et al. (2014) analysed a decade of Ice-Tethered Profiler (ITP; Krishfield et al., 2008; Toole et al., 2011) data (2004–2015), identifying 127 eddies, of which only 39 contained Eurasian Basin water, and most of these were not observed in the Amundsen



Basin. This highlights a significant observational gap regarding the occurrence and properties of IHEs in the Amundsen Basin region.

High-resolution modelling studies, such as those by Müller et al. (2024) and Li et al. (2024), using kilometre-scale simulations (e.g. FESOM2, Danilov et al., 2017), suggest that the Eurasian Basin is densely populated by mesoscale eddies, with eddy
activity closely linked to baroclinic instability of the Atlantic Water boundary current and sea ice dynamics. Complementing these model-based insights, Kuznetsov et al. (2024) reconstructed the ocean state from MOSAiC observations, providing a detailed view of subsurface dynamics and identifying numerous cyclonic and anticyclonic eddies beneath the ice, most of which appear to be in a quasi-steady state. Future model studies would strongly benefit from an eddy inventory in the Eurasian basin, as models need to be validated with observational data.

This paper aims to provide a detailed characterisation of wintertime intrahalocline (sub)mesoscale eddies in the Amundsen Basin, using MOSAiC hydrographic and velocity data to investigate their dynamics, thermohaline properties, formation processes, and variability among individual eddies.

## 2  Methods

### 2.1  Data

The data used in this study were collected during the MOSAiC expedition (Nicolaus et al., 2022; Rabe et al., 2022). In particular, we use data from the Distributed Network (DN) (Rabe et al., 2024) installed around the Central Observatory (CO), where the Polarstern was anchored to the ice, and restrict our analysis to the winter season (October 19 2019, until March 15 2020). This DN was an arrangement of different autonomous ice-tethered systems that aimed to collect various Arctic Ocean properties at different temporal and spatial scales. The DN includes both fixed-depth time series data and vertical profiles.
Although fixed-depth sensors provide high temporal resolution data (on the order of minutes) and capture eddy signatures (Hoppmann et al., 2022), they are unsuitable for detailed characterisation of individual eddies. This is because fixed-depth data do not capture the full vertical structure of IHEs, which require vertical profiling to resolve their thermohaline and velocity structure. Therefore, this study exploits vertical profile data exclusively to analyse the structure and dynamics of wintertime IHEs. We focus on the three instrument deployment locations, termed L-sites, which were positioned at a distance of approximately 12–24 km around the CO (Figure 1b). At the L-sites, Ice-Tethered Profilers (ITPs) (Krishfield et al., 2008; Toole et al.,
2011) provided Conductivity, Temperature and Depth (CTD) measurements and Autonomous Ocean Flux Buoys (AOFBs) equipped with Acoustic Doppler Current Profilers (ADCP) (Stanton et al., 2012) measured horizontal velocity. We also used CTD measurements from the surface to the ocean floor and velocity data from the shipboard ADCP conducted at the CO from the Polarstern.

The three ITPs deployed at the L-sites of the DN conducted different acquisition times and with different profiling times and vertical extensions (Table 1). The L1 site had a profiling cycle consisting of an 18-hour interval followed by a 6-hour interval; L2 followed a more complex cycle of 36, 6, 24, and 6-hour intervals; and L3 had the highest temporal resolution, recording 8 profiles per day. Given the drift velocity of the sea ice, we can compute the distance between the profiles, which ranged between



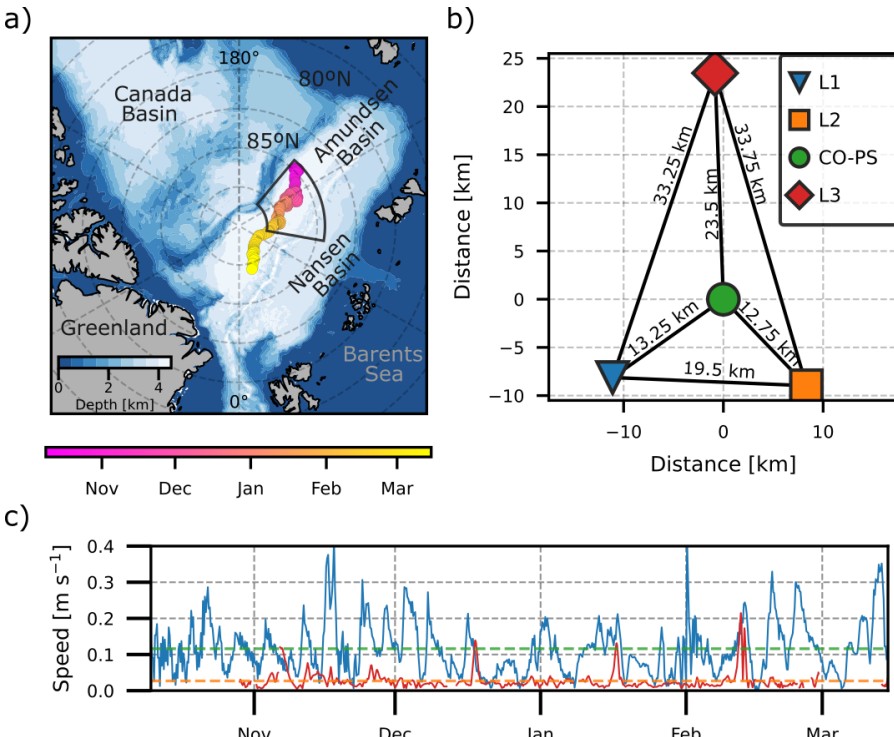

**Figure 1.** a) Drift track of the Central Observatory (CO) on the MOSAiC expedition from October 9 2019 to March 15 2020. The blue scale represents the bathymetry ($\times 10^3$ m). The black polygon in panel a shows the main area of study. b) Spatial position of the L sites around the CO on October 10 2019. c) Drift speed of the CO (blue line) and the mean current speed averaged from the mixed layer to 100 m (red line). The dashed green and orange lines show the mean drift and mean current speed, respectively.

1 and 10 km, where L3 has the highest horizontal resolution and L2 the lowest. The ITPs installed at L1 and L2 survived the
whole winter and drifted until the Fram Strait. Unfortunately, after a ridging event, the L3 ITP stopped measuring on January
31, 2020.

The zonal and meridional components of the absolute velocity profiles were measured using an ADCP with a vertical range
from 10 to 80 meters, a vertical resolution of 2 meters, and a 2-hour sample interval. The L3 ADCP stopped measuring at
the same time as the L3 ITP. Unfortunately, the L2 ADCP did not sampled since the beginning due to technical problems.
However, as an alternative, we used data from the Polarstern's shipboard ADCP in cases where L2 and the CO were aligned.

## 2.2   Eddy detection

In this study, the DN moved with the sea-ice at a mean drift speed of 0.11 m s$^{-1}$, while the underlying ocean current below
the mixed layer had an average speed of 0.02 m s$^{-1}$ (Figure 1c). Because the ice drift is an order of magnitude faster than the
ocean current, the ice-tethered platforms effectively move quickly relative to the ocean features beneath. This large difference





**Table 1.** List of instrumentation, site names, period of measurement, temporal and vertical resolution deployed in the Distributed Network of the MOSAiC expedition used in this study.

| Site | Buoy | Instrument | Time period | Profiling frequency [h] | Depth range [m] | Vertical res. [m] |
|------|------|-----------|-------------|------------------------|-----------------|-------------------|
| L1 | ITP 111 | CTD | 07-10-2019 – 11-06-2020 | 6-18 | 10-200 | 1 |
| L1 | AOFB | ADCP | 07-10-2019 – 27-02-2020 | 3 | 12-70 | 2 |
| L2 | ITP 94 | CTD | 08-10-2019 – 29-07-2020 | 6-24-6-36 | 10-200 | 1 |
| CO-PS | AOFB | ADCP | 14-10-2019 – 19-03-2020 | 2 | 12-70 | - |
| L3 | ITP 102 | CTD | 11-10-2019 – 31-01-2020 | 3 | 10-200 | 1 |
| L3 | AOFB | ADCP | 10-10-2019 – 22-01-2020 | 2 | 10-200 | 2 |
| CO-PS | Polarstern | CTD | 14-11-2019 – 02-20-2020 | - | 1-4000 (Bottom) | 1 |
| CO-PS | Polarstern | ADCP | 28-10-2019 – 04-06-2020 | 1/60 | 25-200 | 8 |

in speeds justifies the quasi-synoptic assumption, which means that measurements from the ice-advected platforms can be considered as near-instantaneous snapshots ("frozen fields") of the slower-evolving ocean eddies (Manley and Hunkins, 1985; Krishfield et al., 2008). This interpretation agrees with observations that Arctic eddies propagate at speeds roughly an order of magnitude slower than the sea ice drift (von Appen et al., 2018).

     To identify eddies, we follow the methodology suggested by Timmermans et al. (2008) and Zhao et al. (2014). First, we

visually identify maximum isopycnal tilting (vertical displacement) anomalies using CTD vertical casts from the ITPs. Anticyclonic eddies are identified by convex-shaped isopycnal displacements in the density sections, whereas concave displacements indicate cyclonic eddies. Second, we analyse the velocity profiles measured by the ADCPs, looking for a speed anomaly characterised by two local maxima in horizontal speed on either side of the isopycnal displacement centre; thus, at least two profiles must be sampled within this speed anomaly. Timmermans et al. (2008) only considered eddies sampled with at least four pro-

files; however, since we have additional ADCP data, we relax this criterion to require at least two profiles, acknowledging that this may limit the precise characterisation of the eddy centre. Figure 2 shows two examples of eddies: E8 (a), detected at L3 with the highest horizontal resolution, and E9 (b), observed at L2 with the lowest resolution. These cases illustrate that relaxing the four-profile criterion is justified when velocity data are available to identify an eddy (see Table 2 for details).

     In general, the eddy core is (close to) a solid body rotation and its outer edge defines the radius of maximum velocity

(Chelton et al., 2011). Once the eddy-like structures have been identified, we need to corroborate that the velocity follows a solid-body rotation (Nurser and Bacon, 2014) to confirm the dynamics are in agreement with eddies and not other features such as meanders or fronts. As a pre-processing step, the velocity is rotated by $\theta$, which is the angle between two consecutive measurements along the drifting pathway, such as that $v_r$ is the along-track drift velocity and $v_\theta$ is the cross-track drift velocity (eddy's azimuthal velocity):





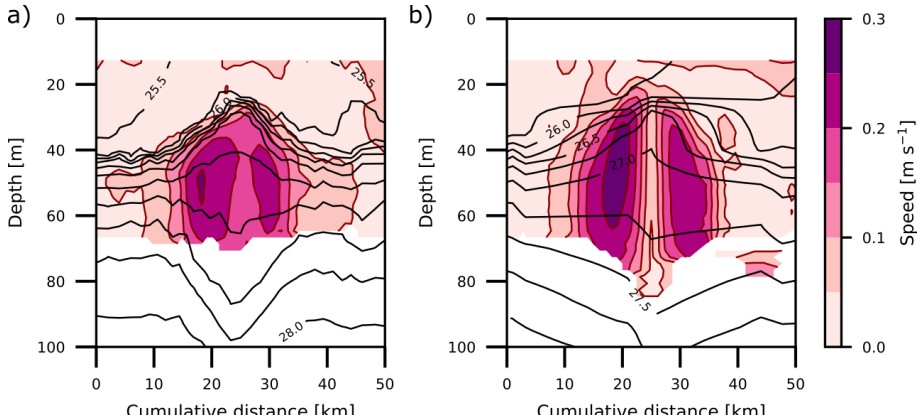

**Figure 2.** Cumulative distance–speed sections for (a) eddy E8 (January 15 2020) and (b) eddy E9 (February 12 2020). Black contours indicate isopycnals spaced every 0.25 kg m$^{-3}$.

$$v_r = u\cos(\theta) + v\sin(\theta) \tag{1}$$

$$v_\theta = -u\sin(\theta) + v\cos(\theta) \tag{2}$$

$$\theta = \arctan\left(\frac{y_2 - y_1}{x_2 - x_1}\right) \tag{3}$$

where $(x_1, y_1)$ is the position of the first measurement and $(x_2, y_2)$ of the second measurement. $v_\theta$, also termed the swirl velocity, provides a sense of the rotation of the fluid (i.e., the tangential velocity component within a swirling flow).

In the theoretical Rankine vortex (Acheson, 1990), the azimuthal velocity increases linearly with the distance to the centre, having the maximum value of the velocity at $R_{\max}$; the distance between the location of the absolute smallest swirl velocity (centre of the eddy) and the maximum azimuthal velocity ($V_{\max}$) (Figure 3, red line). A method for computing the azimuthal velocity $v_\theta$ of a theoretical eddy as a function of its radius (Equation, 4) was presented by Castelão and Johns (2011), based on a study of Gulf Stream rings (Olson, 1980, 1991):

$$v_\theta(r) = \begin{cases} r\dfrac{V_{\max}}{R_{\max}}, & \text{for } r \leq R_{\max} \\[2mm] V_{\max}\exp\left(-\dfrac{r - R_{\max}}{\lambda}\right), & \text{for } r > R_{\max} \end{cases} \tag{4}$$

where $\lambda$ is a damping coefficient that indicates decay. Equation 4 assumes an inner part of the eddy that rotates like a solid body ($r \leq R_{\max}$) and an outer part ($r > R_{\max}$) where the velocity decays rapidly at the e folding scale $\lambda$, which is typically about $\frac{1}{3}R_{\max}$. Here, we are focusing on the inner part (i.e., the core of the eddy). Furthermore, the dynamics outside the limit of $R_{\max}$ are out of the scope of this study.





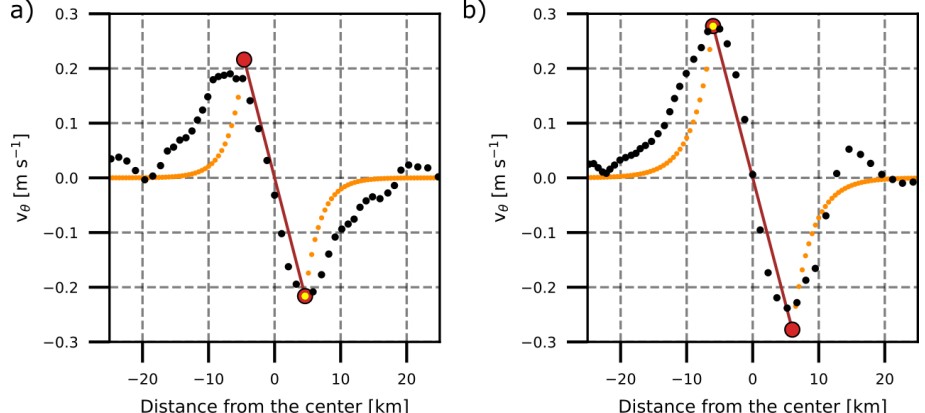

**Figure 3.** Azimuthal velocity ($v_\theta$, ms$^{-1}$) (a) eddy E8 (January 15 2020) and (b) eddy E9 (February 12 2020). Black dots show the measured $v_\theta$ values at the depth level of maximum azimuthal velocity from ADCP data, computed using Equation 1. The solid red line represents the inner part of the Rankine vortex model (Equation 4, $r \leq R_{\max}$), while yellow dots represent the outer part (Equation 4, $r > R_{\max}$). The maximum azimuthal velocity ($V_{max}$) used in the model is marked by a red dot with a yellow centre.

The azimuthal velocity ($v_\theta$) calculated using Equation 1 was compared to the theoretical Rankine vortex (Equation 4) to assess whether the eddy cores exhibit solid-body rotation. In both E8 (January 15) and E9 (February 12) (Figure 3), the observed azimuthal velocity profiles measured by the ADCP closely follow the theoretical shape. The inner region displays solid-body rotation, while the outer region shows a rapid velocity decay, consistent with the Rankine vortex structure.

## 2.3    Determining the centre of the eddy and its radius

Several methods have been used to determine the centre of eddies in the open ocean. However, most need a surface expression of the eddy to obtain the horizontal velocity field (Chelton et al., 2011). Our study region is covered by thick sea ice, making detecting eddies using remote sensing data impossible. Thus, to estimate the centre of the eddy and obtain an accurate approximation of its radius, we applied the Maximum Swirl Velocity (MSV) method as described by Castelão et al. (2013). This method assumes that an eddy is axisymmetric, with all the momentum associated with the azimuthal component of the velocity.

Hence, its centre is defined as the reference point in a cylindrical coordinate system that maximises the measured azimuthal velocity $v_\theta$ among the available data points, while the radial component of the velocity $v_r$ is vanishing $V = \sqrt{v_\theta^2 + v_r^2} = v_\theta$.

To find the centre, we followed the approximation of Nencioli et al. (2008). First, we defined an area of $2R_{max} \times 2R_{max}$ around the location where the minimum velocity inside the eddy was measured, dividing it into a 100 m resolution grid. We then used every point of the grid as a theoretical centre, decomposing all of the ADCP velocities, as in Equation 1, into

tangential and radial components relative to each candidate centre. As $v_\theta$ has opposite signs for cyclones and anticyclones, it is easier to determine the centre of the eddy by finding the location where $v_r$ is minimal, thereby minimising the cost function





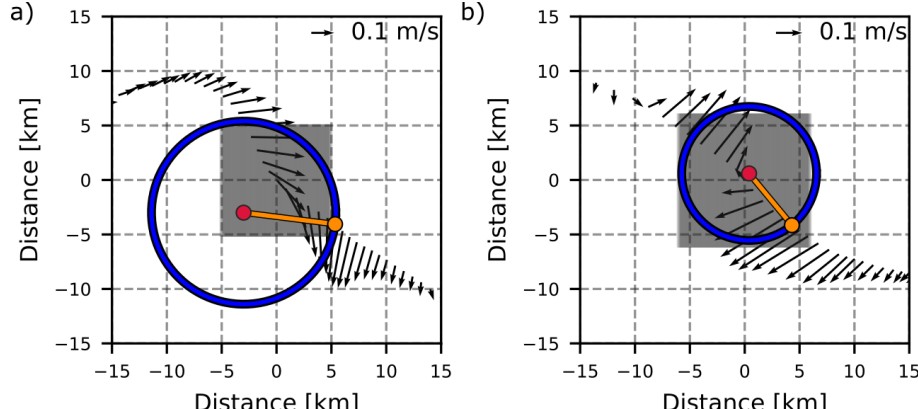

**Figure 4.** Velocity vectors of the eddies in (a) eddy E8 (January 15 2020) and (b) eddy E9 (February 12 2020) at the depth of maximum velocity. The grey area shows the grid used for the detection of the eddy centre, the red dot shows the estimated eddy centre using the methodology of Nencioli et al. (2008), the orange dot is the maximum azimuthal velocity ($V_{max}$) location and the orange line show the distance between the location of the absolute smallest azimuthal velocity (centre of the eddy) and $V_{max}$. The blue circle marks the inner part of the eddy.

$J$ (Castelão et al., 2013):

$$J = \frac{1}{2N} \sum_{n=1}^{N} \left( \frac{v_{rn}}{V_n} \right)^2 \qquad (5)$$

where $N$ is the number of ADCP measurements used and $V$ is the speed. Once the centre is detected, we recalculated the radius $R_{max}$ as the distance between the theoretical centre and $V_{max}$ (as shown in Figure 4).

We apply this eddy detection method to eddies E8 and E9 (Figure 4). The centre of the eddy E8, calculated by minimising the cost function $J$, lies approximately $5\,\mathrm{km}$ from the point of minimum velocity in the transect, and its radius is estimated at $8.4\,\mathrm{km}$. In contrast, the radius obtained by measuring the distance between the locations of minimum and maximum velocity along the same transect is considerably smaller, about $4.6\,\mathrm{km}$. For eddy E9, the L2 transect crossed nearly through its centre, and the difference between the two radius estimates is minimal ($5.9\,\mathrm{km}$ - $6.1\,\mathrm{km}$).

## 2.4 Calculation of Rossby radius

The Rossby radius of deformation is a fundamental scale in geophysical fluid dynamics that characterises the horizontal extent over which baroclinic processes, such as eddies, are influenced by the Earth's rotation. It represents the length scale at which the restoring force due to stratification (buoyancy) is balanced by the Coriolis force (rotation), and is thus a critical parameter in controlling the dynamics of mesoscale structures (Nurser and Bacon, 2014). To constrain the local scale of these mesoscale processes, we calculated the first ($L_1$) mode of the baroclinic Rossby radius of deformation using the approximation of Wang et al. (2013) (Equation 6):



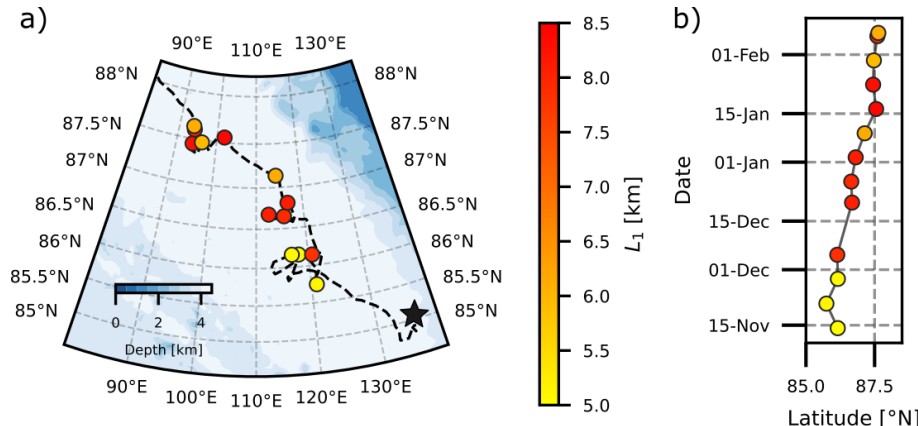

**Figure 5.** The first baroclinic Rossby radius of deformation mode ($L_1$) calculated from CTD vertical casts obtained aboard RV Polarstern during the MOSAiC drift (dashed black line), with the start of the drift (9 October 2019) marked by a black star. Spatial distribution of $L_1$ is shown in (a), and the time series of $L_1$ as a function of latitude is shown in (b).

$$\frac{\partial}{\partial z}\left(\frac{f^2}{N^2}\frac{\partial F_m}{\partial z}\right) = -\frac{1}{L_m^2}F_m, \tag{6}$$

where $N$, the Brunt–Väisälä frequency, is derived from the CTD vertical cast. $f$ is the Coriolis parameter ($f = 1.45 s^{-4}$),

$F_m$ is the dynamic mode eigenfunction and $L_m$ is the Rossby radius of the $m^{th}$ baroclinic mode. To solve this equation, we apply a flat-bottom boundary condition ($\frac{dF_m}{dz} = 0$ at $z = 0, -H$), appropriate for our study region within the deep Amundsen Basin. This area is distant from major bathymetric features, such as ridges or seamounts, and can be approximated as laterally homogeneous and vertically unaffected by topographic constraints.

     In Figure 5, we show the spatial distribution of $L_1$ in our study area. Stations located close to each other sometimes show

different $L_1$ values, likely due to local variations in the vertical stratification of the water column. These differences reflect the sensitivity of the method to small-scale changes in water column stability, which are captured in the CTD profiles. The mean value of $L_1$ in the study area is $6.93$ km, which justifies our use of the term (sub)mesoscale throughout the paper, as several eddy structures observed fall near or below this threshold.

## 3    Results

### 3.1    Examples of two characteristic eddies

We observed 9 eddies, which, for the purpose of explanation, we label E1 to E9. We start this section by analysing in detail two representative anticyclonic eddies: E8, observed on January 15, and E9, observed on February 12 (see Table 2). These examples are used throughout the Methods section to illustrate our detection and characterisation approach. The E8 eddy



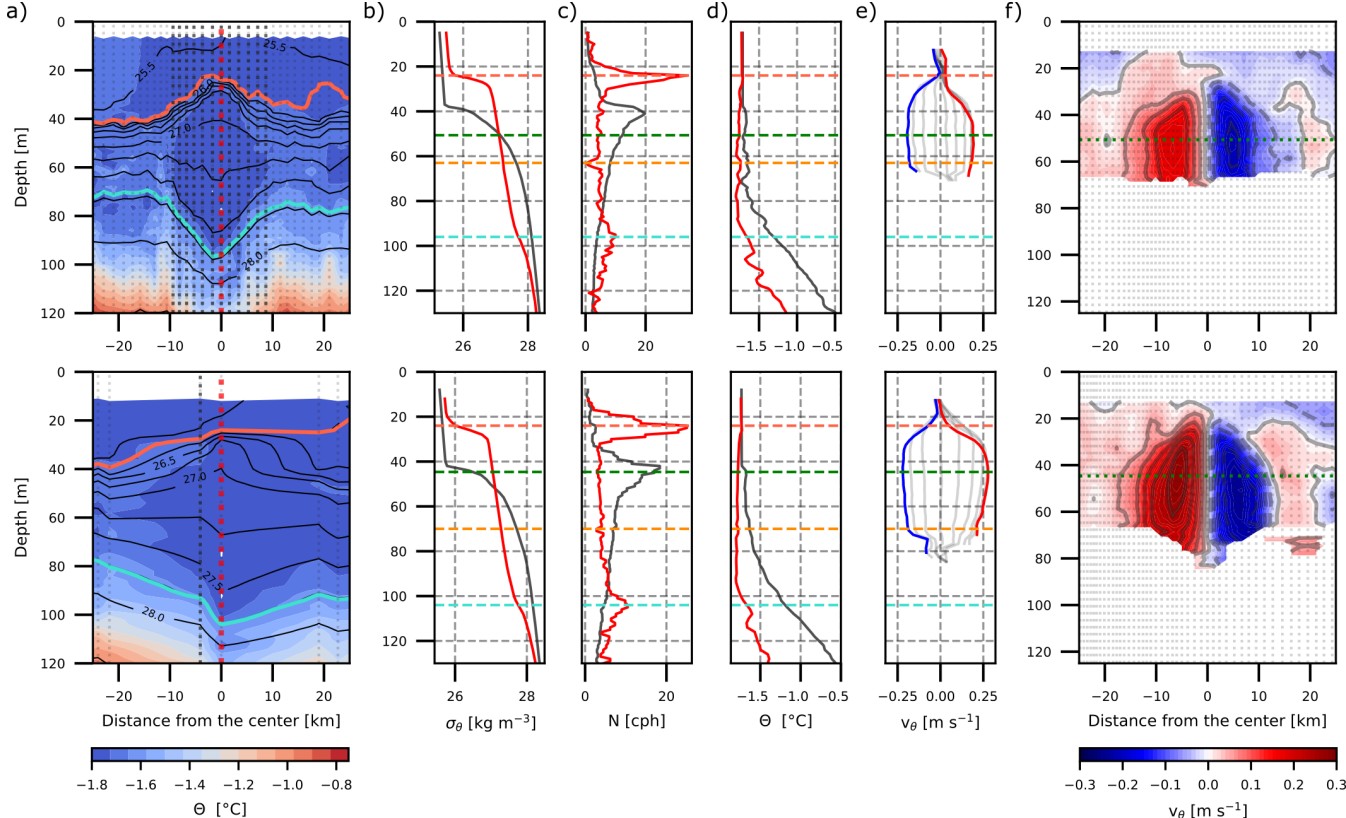

**Figure 6.** Details of the eddies E8 (upper panel) and E9 (lower panel). a) Cross sections of conservative temperature (Θ) with isopycnals shown as black contours spaced every 0.25 kg m$^{-3}$; red and cyan contours indicate the upper and lower limits of the eddies, respectively, and the dashed vertical red line marks the central eddy profile. Vertical profiles of density ($\sigma_\theta$) (b), buoyancy frequency (N) (c), conservative temperature (Θ)(d), and azimuthal velocity ($v_\theta$) (e). The red line shows the central profile, and the grey line shows the mean profiles at $\pm$ 30 km around the eddy. Dashed horizontal lines show the top (red), the bottom (cyan), the maximum azimuthal velocity level (green) and the eddy core depth (orange). (f) Cross section of azimuthal velocity $v_\theta$ with velocity contours in grey every 0.05 ms$^{-1}$. The green dotted line indicates the depth of maximum velocity. The dotted vertical light grey lines in (a) and (f) marker the measurement profiles, in (a) the darker lines are the profiles inside the eddy.

(Figure 6, upper panels) was captured by the L3 buoys with 12 ITP profiles within the solid-body rotation region (core) and 10

ADCP profiles in the inner core. The MSV method (see section 2.3) revealed a radius of 8.4 km. The E9 eddy (Figure 6, lower panels), captured by the L2 buoy with only 2 ITP profiles and 9 ADCP profiles, had a radius of 6.14 km. We chose these two eddies as examples because they were the largest and best-sampled eddies detected along the track. Although the CTD profiles only coarsely resolved E9, the ADCP data did resolve it well, and it suggests the eddy was crossed almost through its centre (Figure 4).



To characterise the eddies, we used profiles of conservative temperature ($\Theta$), absolute salinity ($S_{A_C}$), density ($\sigma_\theta$), buoyancy frequency ($N$), and azimuthal velocity ($v_\theta$). We selected the central profile, where the isopycnal displacement was greatest—based on ITP data (Figure 6a, red dashed line). The $N$ profiles were used to define the vertical boundaries of the eddies. We define the upper limit of the eddy (Figure 6, red line) as the depth of the first peak in $N$, and the lower limit (Figure 6, cyan line) as the depth of the second peak in $N$, and the core depth is defined as the depth where $N$ reaches a minimum between

these two limits. Our definition of the core depth differs from that used by Timmermans et al. (2008), who used the level of minimum $\Theta$. That criterion was not applicable to the eddies observed in this study, as no clear $\Theta$ minimum was present. The eddy thickness is thus given by the depth difference between its upper and lower boundaries. The eddies observed are IHEs, located near the base of the mixed layer and interacting with the upper halocline. As they translate, they uplift the mixed layer, making it thinner. The eddy E8 yielded a decrease in the MLD from 41 m to 24 m from the eddy edge to the eddy centre,

which is similar to that resulting from the eddy E9, with a decrease of 20 m depth. At the upper boundary, the isopycnal was displaced upwards 14 m in both eddies; at the lower boundary, it was displaced downwards by 23 m in E8 and 19 m in E9. In E8, the eddy's upper boundary is located at depth with $v_\theta \approx 0$ (Figure 6d), indicating good agreement between the CTD and ADCP data. In E9, the eddy's upper boundary, as determined using the $N$ profiles, does not align with the depth with $v_\theta \approx 0$, likely due to the coarser temporal resolution of the CTD data. The maximum azimuthal velocity of the eddy was 0.25 ms$^{-1}$

and 0.28 ms$^{-1}$ in E8 and E9, respectively.

### 3.2    Properties of all eddies detected during the winter season

In the period from October 9 2019 to March 15 2020, we detected nine well-developed anticyclonic eddies in the central part of the Amundsen Basin (Figure 7a, Table 2). These eddies are consistent with our criteria, showing evident isopycnal displacement and solid-body rotation. At the L3 site, we sampled five eddies at high horizontal resolution, enabled by the

high-frequency sampling of both ITP and ADCP instruments at this location (Figure 7, diamond markers), and their presence is evident in both the isopycnal displacement and the large subsurface azimuthal velocity. The ITP at the L2 site had a more complex profiling schedule (Table 1), with a greater horizontal distance between consecutive profiles, making the identification of eddies by isopycnal displacement alone more challenging. However, when analysing the ADCP data, we detected one eddy in December and one in February (Figure 7, square markers). We do not have hydrographic data for the inner part of the eddy

in December (E7), and only two profiles are available for the eddy in February (E9). At the L1 site, we detected two eddies, one in October (E3) and one in November (E4) (Figure 7, inverted triangle markers). There is a one-month gap in eddy detections in the DN (December 17, 2019 – January 15, 2020). However, there were no obvious changes in the drift speed of the DN during that period (Figure 1c, mean speed of $\sim$0.1 ms$^{-1}$). Hence, the absence of eddies is likely unrelated to the temporal and spatial resolution of the measurements.

We now describe the mean properties of the detected eddies (Figure 7 and Table 2). The dynamical nature of the eddies can be characterised through the interplay between four key parameters: the eddy radii estimated from the MSV ($R_m = 6.09 \pm 1.4$ km), the first mode of the baroclinic Rossby radius of deformation ($L_1 = 6.9 \pm 1.3$ km), the maximum azimuthal velocity ($V_{max} = 0.14 \pm 0.07$ ms$^{-1}$), and the Rossby number ($R_o = 0.32 \pm 0.14$). The condition $R_m < L_1$ places these eddies in





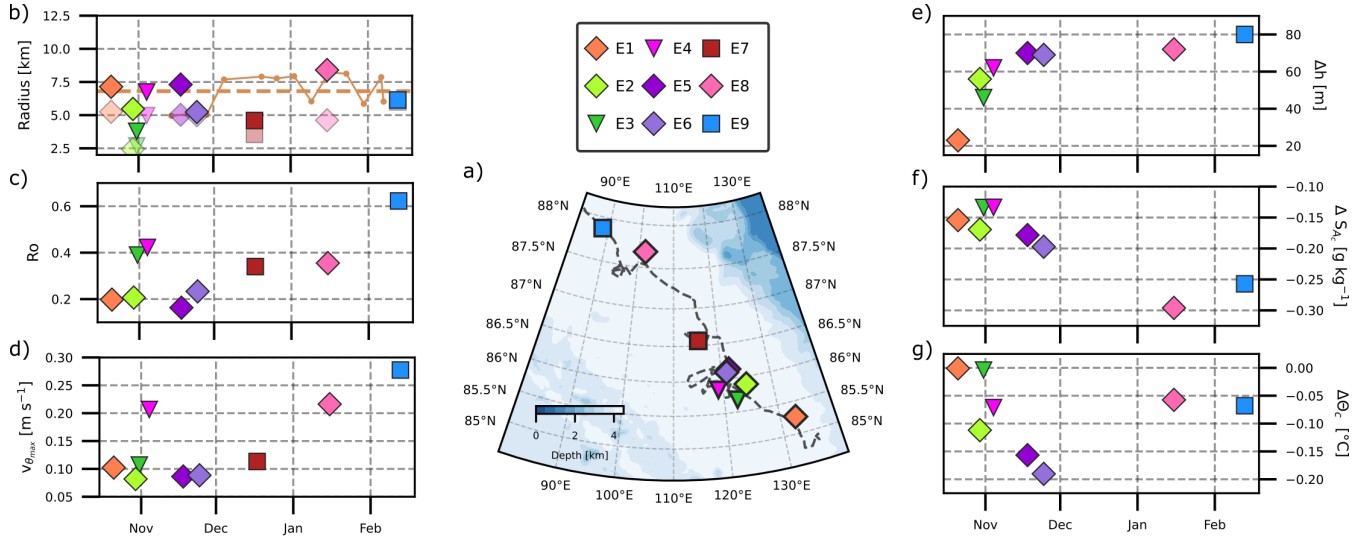

**Figure 7.** Properties of the nine eddies (E1–E9) detected during the winter 2019–2020 MOSAiC drift and their spatial locations (a). Markers indicate the L-site of detection (inverted triangles: L1, squares: L2, diamonds: L3). Time series panels show the eddy radius (b), Rossby number (Ro) (c) and maximum azimuthal velocity ($V_{\theta_{max}}$) (d) on the left; and eddy thickness ($\Delta h$) (e), core absolute salinity anomaly ($\Delta S_{A_C}$) (f) and core conservative temperature anomaly ($\Delta \Theta_C$) (g) on the right. In (b), we compare the radius estimated from the distance between $V_{\theta_{max}}$ and the centre located in the buoy drift (Figure 3, translucent markers) with that calculated using the MSV method (Figure 4). The first mode of the Rossby radius $L_1$ is shown in orange, with mean values indicated by dashed lines in the same colour.

the (sub)mesoscale regime, indicating a transitional dynamical scale. We computed $R_o$ using the cylindrical approximation

$R_o = \frac{2U}{fR}$ (Zhao et al., 2014), where $U$ is $V_{max}$, $f$ is the Coriolis parameter and $R$ is the radius. This yields $R_o$ ($0.16 < R_o <$ 0.62) that is consistent with the quasi-geostrophic balance. It is interesting that the radii would be underestimated by $\approx 1.7\,\mathrm{km}$ if we did not perform the MSV correction (Figure 7b), which would indicate that the eddies appear closer to the submesoscale regime than they actually are. In the centre of the eddies, the MLD becomes shallower on average by $11.5 \pm 5.73\,\mathrm{m}$, ranging from 33.6 to 22.12 m depth. The eddy thickness differs by 23 to 80 m, having an average thickness of $59.75 \pm 18.1\,\mathrm{m}$. The

depth of the eddy centre ($D_c$) were found at $59.3 \pm 13.6\,\mathrm{m}$ depth with an average temperature of $\Theta_c = -1.773 \pm 0.04°\mathrm{C}$, salinity of $S_{A_c} = 33.954 \pm 0.26$ and a potential density of $\sigma_{\theta_c} = 27.49 \pm 0.24\,\mathrm{kg\,m^{-3}}$. These values correspond to the range of surface waters in the Amundsen Basin, but then, if we look at the anomalies against the mean values of the profiles at $\pm 30\,\mathrm{km}$ around the eddy, we find small yet significant anomalies ($\Delta\Theta_c = -0.082°\mathrm{C}$, $\Delta S_{A_c} = -0.189\,\mathrm{g\,kg^{-1}}$, $\Delta\sigma_{\theta_c} = -0.15\,\mathrm{kg\,m^{-3}}$), that we will use later to discuss the possible origin of the eddies.

In the Arctic Ocean, density is primarily driven by salinity changes rather than temperature due to the well-developed halocline, as cold waters remain close to the freezing point, minimising thermal effects (Aagaard and Carmack, 1989; Carmack et al., 2016). This dependency is evident in the core properties, with density anomalies linearly associated with salinity anomalies (Figure 8a). A similar behaviour is observed when we compare $V_{max}$ with the thickness (Figure 8c), which is also expected,



**Table 2.** Summary of all detected IHE eddies during the winter MOSAiC drift. Eddies are labelled sequentially (E1 to E10) based on their chronological order of detection. Site, date, and core properties are listed for reference. Mixed layer depth in the central profile/mean water state $\Delta MLD$ [m], Thickness $\Delta h$ [m], Core depth $D_c$ [m], Core values of conservative temperature $\Theta_c$ [°C], absolute salinity $S_{A_c}$ [ g kg$^{-1}$], density $\sigma_{\theta_c}$ [kg m$^{-3}$] and their anomaly $\Delta$ values. Rossby number $R_o$, maximum azimuthal velocity $V_{max}$ [ m s$^{-1}$] and depth $D_{V_m}$.

| Eddy | Site | Date | $\Delta MLD$ | $\Delta h$ | $D_c$ | $\Theta_c$ | $S_{A_c}$ | $\sigma_{\theta_c}$ | $R_m$ | $V_{max}$ | $D_{V_m}$ | $Ro$ | $\Delta\Theta_c$ | $\Delta S_{A_c}$ | $\Delta\sigma_{\theta_c}$ |
|------|------|------|------|------|------|------|------|------|------|------|------|------|------|------|------|
| E1 | L3 | 21-10 | 22/31 | 23 | 36 | -1.72 | 33.84 | 27.28 | 7.14 | 0.10 | 33 | 0.19 | -0.001 | -0.154 | -0.124 |
| E2 | L3 | 29-10 | 19/29 | 56 | 73 | -1.71 | 34.26 | 27.8 | 5.46 | 0.08 | 33 | 0.2 | -0.111 | -0.169 | -0.133 |
| E3 | L1 | 31-10 | 28/32 | 46 | 54 | -1.78 | 34.09 | 27.57 | 3.78 | 0.1 | 45 | 0.38 | -0.003 | -0.133 | -0.107 |
| E4 | L1 | 04-11 | 16/35 | 62 | 43 | -1.8 | 33.81 | 27.29 | 6.74 | 0.2 | 37 | 0.42 | -0.071 | -0.133 | -0.105 |
| E5 | L3 | 17-11 | 23/29 | 70 | 69 | -1.81 | 34.18 | 27.72 | 7.3 | 0.08 | 65 | 0.16 | -0.156 | -0.178 | -0.139 |
| E6 | L3 | 24-11 | 21/30 | 69 | 67 | -1.83 | 34.19 | 27.72 | 5.23 | 0.08 | 63 | 0.23 | -0.19 | -0.197 | -0.153 |
| E7 | L2 | 17-12 | - | - | - | - | - | - | 4.59 | 0.11 | 47 | 0.34 | - | - | - |
| E8 | L3 | 15-01 | 24/41 | 72 | 63 | -1.74 | 33.62 | 27.23 | 8.4 | 0.21 | 49 | 0.35 | -0.057 | -0.296 | -0.237 |
| E9 | L2 | 12-02 | 24/42 | 80 | 70 | -1.77 | 33.63 | 27.27 | 6.14 | 0.27 | 45 | 0.62 | -0.067 | -0.257 | -0.205 |

since both of the properties must be in balance to conserve potential vorticity (Cushman-Roisin and Beckers, 2011). In addition
to being thicker, we found that the more energetic eddies also have greater anomalies in the core. A group of eddies is observed
at $R_o$=0.2 (Figure 8b), which is closer to geostrophic balance. This is in contrast to those that tend to higher values of $R_o$,
for which the Coriolis force alone is insufficient to balance the pressure gradient. In such cases, the centrifugal force becomes
dynamically relevant, suggesting a cyclogeostrophic balance —where both Coriolis and centrifugal forces act to balance the
pressure gradient— similar to the findings of Zhao et al. (2014). We find that most of the eddies with larger thickness are
located deeper in the water column (Figure 7d).

No eddies were detected after February 12, the date of the last confirmed detection. Although the DN continued drifting,
the velocity measurements became progressively limited due to the sequential failure of the AOFBs, first at L1 (February
27) and later at the Central Observatory (March 19), leaving the sADCP as the only velocity source, which did not reveal
any coherent eddy signatures. Meanwhile, the ITPs at L1 and L2 remained operational but did not register additional eddies.
By mid-March, after the drift crossed the Gakkel Ridge and transitioned to the Nansen Basin, the mixed layer had deepened
markedly (exceeding 150 m), which likely inhibited the detection of IHEs within the 200 m vertical range of the remaining
ITPs.





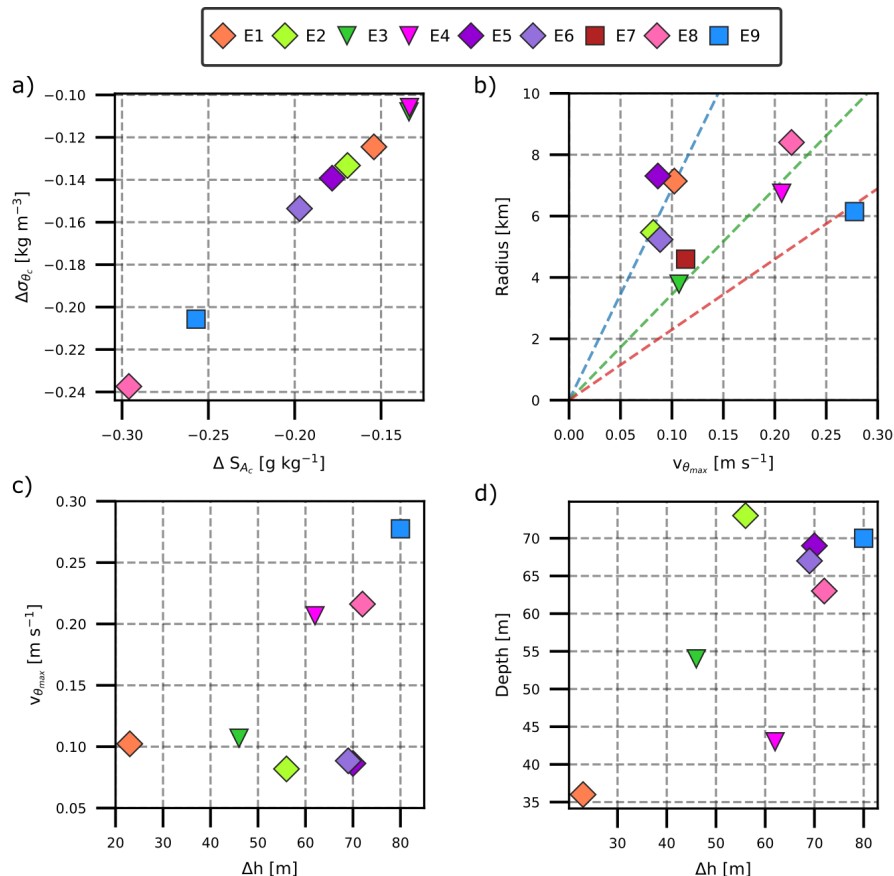

**Figure 8.** Scatter plots of: a) density anomaly ($\Delta\sigma_{\theta_c}$) versus salinity anomaly ($\Delta S_{A_c}$), b) Radius versus maximum azimuthal velocity ($V_{max}$) with dash lines showing $Ro$ of 0.2 (blue), 0.4 (green),and 0.6 (red), c) maximum azimuthal velocity ($V_{max}$) versus thickness ($\Delta h$) and d) Core depth ($D_c$) versus thickness ($\Delta h$). Colours and symbols are as in Figure 7.

## 4 Discussion

### 4.1 Detection of duplicate eddies

The study of eddies under sea ice prompts the question of whether the same eddy has been sampled several times. The answer to this question is not trivial; all the different measurements need to be assessed to constrain it. First, the assumption of quasi-synoptic conditions adopted in section 2.2 implies that the eddies cannot move fast enough to pass several sites within the time frame of a single observation period. A second aspect to consider is the eddy size. Most of the eddies observed by Zhao et al. (2014) in the Amundsen Basin had radii of ≈5 km, consistent with our observations, and the distance between L1 and L3 is

larger than 30 km, making it unlikely that the same eddy would be detected at both locations. However, at shorter distances, such as between L2 and CO, the possibility increases. In particular, eddies were detected at both sites around the exact dates,





**Figure 9.** Drift pathways and cross-track velocity sections for the L-sites from October 29 to November 29. Panels (a), (b), and (c) correspond to sites L3, CO (Polarstern), and L1, respectively. Each panel includes a map on the left showing the drift trajectory of the corresponding site (in dark grey), with the starting point marked by a red symbol and the endpoint by a blue one. The trajectories of the other two sites are shown in light grey for reference. Eddy locations are represented as coloured circles, scaled by their estimated radius and matching the colours used in the left panels. Circles are shown in full colour when the eddy was sampled by the site, and translucent otherwise. The right-hand panels display the cross-track velocity along each site's drift path. Eddies whose cores were crossed are marked with symbols at the top of the panels. When a site passed near the location of a previously detected eddy, this is indicated by a solid-colour segment along the drift path (left panel) and by dashed vertical lines of the same colour in the velocity section (right panel).

and given the prevailing northeastward/southeastward drift, it is plausible that different buoys could have sampled different parts of the same eddy. We investigate this possibility in more detail below.

Between October 29 and November 29, 2019, five eddies were observed at different L sites within a relatively short time window, raising the possibility that some of these detections correspond to the same eddy sampled at various stages of its path. Although the distances between sites such as L1 and L3 exceed 30 km—well beyond the radius of the eddies in the area of ∼6 km—, the temporal coincidence warrants a closer investigation into whether some of these eddies could have drifted between



nearby sites, particularly those closer together such as L2 and CO. For instance, L1 and L3 detected an eddy within two days (E3 on October 31 and E4 on October 29; Figure 9, green and lime circles). Both fell in the smaller radius range but showed

different core characteristics. The sADCP from the Polarstern (Figure 9b, right panel) shows that at the time of these events delimited by the green dashed line, there is no apparent influence from either eddy, suggesting they are independent. From November 15 to 19, a storm affected the ice drift, increasing the speed up to 0.4 $\mathrm{ms}^{-1}$ (Figure 1c) and changing the drifting direction several times (Figure 9, left panels). As a result, the DN platform sampled some sites more than once. Of the three eddies encountered on November 4 (E4), 17 (E5), and 24 (E6) (Figure 9, pink, lilac and purple circles), E4 stands out as a

well-formed eddy with a strong azimuthal velocity of $V_{max} = 0.20$ $\mathrm{ms}^{-1}$ (Figure 9c, right panel). The periphery of this eddy was also observed at the CO site on November 7 (Figure 9b, right panel, dashed pink line), as confirmed by the drift trajectory of CO passing near the core's edge. Site L3 recorded two eddies within a week, with centres separated by 6 km. This suggests that it was the same eddy, which translated at a speed of 0.01 $\mathrm{ms}^{-1}$, and due to the storm, our instruments sampled it twice. Not only do the spatio-temporal scales support this interpretation, but also the thermohaline and kinematic properties (see Table 2),

with small differences likely resulting from the instruments not crossing exactly the same portion or water masses of the eddy. This explains why the estimated radius differs slightly (7.3–5.2 km). Figure 10e shows that the water masses trapped inside E5 and E6 display the same characteristic properties, reinforcing the interpretation that the same eddy was sampled twice.

## 4.2    Origin and generation of eddies

Unlike temperate seas, the generation and trajectory of eddies cannot be remotely observed beneath Arctic sea ice. Although

the western Nansen Basin shows stronger eddy kinetic energy than the interior Eurasian Basin, weaker and less frequent eddies have also been observed in the central Arctic (Von Appen et al., 2022). Literature shows that most of the efforts to categorise Arctic eddies have focused on the differences in the thermohaline properties of their cores. Based on this, eddies have been classified into Canadian water and Eurasian water eddies. In turn, this classification is divided into shallow (<80 m) and mid-depth (>80 m) core depth, respectively (e.g., Zhao et al., 2014). The nine eddies found in this study are shallow

Eurasian water eddies, containing salty but warmer waters than those studied by Zhao et al. (Figure 10 b). The $\Theta$-$S_A$ diagram in Figure 10 shows three different characteristic shapes: (i) fluctuant temperature with a smooth "wedge" shape in E8 and E9 (Figure 10c), (ii) a smoother curve in October (Figure 10d) and (iii) a prominent "wedge" shape in November (Figure 10e). The smooth curve in Figure 10c is the typical $\Theta$-$S_A$ diagram observed in the surface Amundsen Basin water, with the temperature minimum just above the thermocline (Rudels et al., 1996). It results from advective-convective processes (Steele and Boyd,

1998). Following the formation of the winter mixed layer, fresher water originating from the Russian shelves and transported via the Transpolar Drift reaches the freezing point and becomes dense enough to convectively mix with the existing mixed layer (Kikuchi et al., 2004). This process generates the cold halocline layer, a key feature of the Nansen Basin surface structure. In summer, meltwater from sea ice accumulates at the surface, stratifying above the cold halocline layer. As freezing resumes in early winter, this freshwater cools to the freezing point and begins to convect into the halocline, forming the distinctive $\Theta$–$S_A$

"wedge" shape. Figure 10e shows the $\Theta$-$S_A$ diagram arising from a convective cold halocline, resulting from the stratification of summer sea ice meltwater and the remnants of a winter mixed layer (Steele and Boyd, 1998). It has the particularity of a





prominent wedge, typical of surface conditions in the Nansen Basin during late autumn, when the water column is actively adjusting to the changing surface freshwater input and atmospheric cooling. A similar process occurred inside the eddy (Figure 10c), but the refreezing and convection during winter altered the upper part, making the wedge smoother than in the early

winter season (Kikuchi et al., 2004). Comparing the $\Theta$-$S_A$ diagram of the eddies with the surrounding water, we find that E1 and E2 have trapped similar water masses, suggesting these eddies were likely formed in the same region. The $\Theta$-$S_A$ diagrams of the other eddies have a wedge shape consistent with the typical processes occurring at the surface of the Nansen Basin, which is not seen in the surrounding waters at the time of the observations.

Although the exact generation mechanism of the observed eddies cannot be determined with certainty, the water mass anal-
yses provide valuable insight into the stratification and convective processes that likely precondition the upper water column before eddy formation. In particular, the presence of cold, fresh anomalies and a sharpened halocline in the eddy cores suggests that local convection during winter, possibly associated with lead refreezing, played a role. Additionally, the geographic location of the eddies—well within the Transpolar Drift path—indicates that they may have formed upstream, in regions influenced by freshwater input from the Siberian shelves. This supports the hypothesis that baroclinic instability, facilitated by
strong vertical stratification and preconditioning from prior surface forcing (e.g., convection in leads), is a plausible generation mechanism (Bush and Woods, 2000). Another hypothesis, supported by observations and modelling, suggests that baroclinic instabilities—largely independent from surface conditions due to the persistent stratification—could be the dominant generation mechanism throughout the year (Meneghello et al., 2021). These mechanisms are not mutually exclusive: thermohaline convection in leads may precondition the water column, creating vertical shear and density structures that enable baroclinic
instability. Thus, eddy generation may result from a combination of surface-driven convection and deeper baroclinic adjustment. Given the lack of strong horizontal velocity shear and the observed eddy scales matching the first baroclinic deformation radius, barotropic instability is unlikely to be a dominant mechanism in this region.

## 5   Conclusions

This study presents a detailed characterisation of intrahalocline eddies (IHEs) in the Amundsen Basin, based on hydrographic
and velocity data collected during wintertime in the MOSAiC expedition. Nine well-defined anticyclonic eddies were identified, with radii of $R_m = 6.09 \pm 1.4$ km and thicknesses ranging from 23 to 80 m, all exhibiting solid-body rotation. The thermohaline properties of the water masses trapped within their cores allowed us to infer pre-existing stratification conditions, providing insight into the environmental background from which these eddies formed.

Our results show that IHEs locally alter the vertical stratification, shoaling the mixed layer by over 10 m and affecting
the stability of the halocline. Their horizontal and vertical scales, together with Rossby numbers in the range $0.16 < R_o < 0.62$, place them within a transitional dynamic regime between meso- and submesoscale, consistent with quasi-geostrophic dynamics. Applying the Maximum Swirl Velocity (MSV) method resulted in radius estimates that were on average 1.7 km (25%) larger than those obtained using simpler transect-based methods. This correction is relevant because underestimating





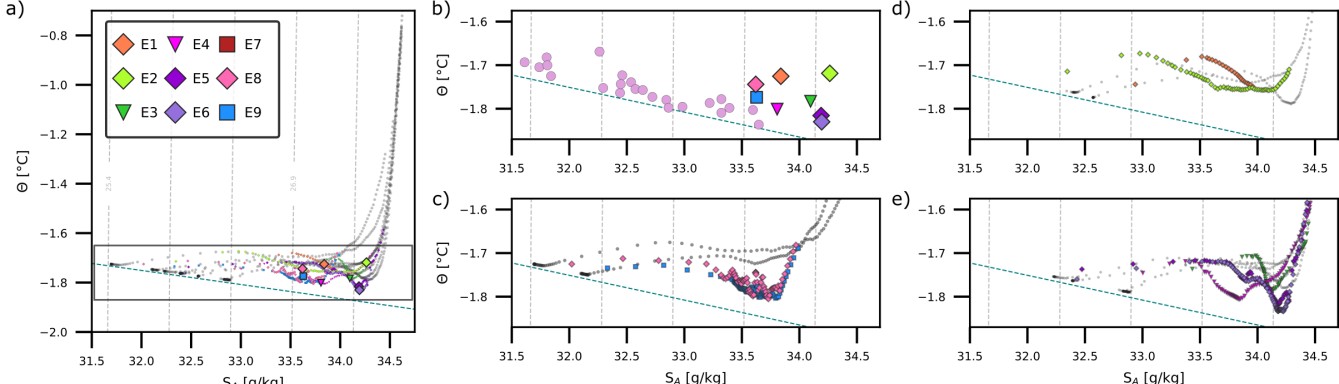

**Figure 10.** $\Theta$-$S_A$ diagrams. Density contours and freezing temperature (at surface pressure) are shown in grey dashed lines and in blue dashed lines, respectively. Mean profiles of the surrounding water are shown in grey, and the colours represent each eddy up to 90 meters depth with the core properties shown by the larger markers. b) shows the core $\Theta$-$S_A$ values, and in pink the core values of Eurasian eddies from Zhao et al. (2014). Groups of eddies with similar $\Theta$-$S_A$ curves indicating different generation processes: c) refreezing and convection, d) advective-convective and e) convective cold halocline.

eddy size can lead to significant misinterpretation of their transport capacity, energy content, and dynamical role, particularly in under-ice conditions where spatial sampling is sparse.

Although the generation mechanisms remain uncertain, the consistent presence of cold and fresh anomalies in the eddy cores suggests that local convection and/or baroclinic instability may play a role in their formation. Future studies should incorporate higher resolution ($\sim 5$ km), spatially distributed autonomous observations capable of resolving the lower end of the mesoscale, in order to advance our understanding of the role of IHEs in central Arctic Ocean dynamics, stratification, and the lateral transport of heat and freshwater.

*Data availability.* All datasets used in this study are publicly available, in compliance with the MOSAiC data policy. CTD Polarstern: Tippenhauer et al. (2023); ITPs: Toole et al. (2011); AOFBs: Stanton and Shaw (2023); ADCP Polarstern: Tippenhauer and Rex (2020)

*Author contributions.*

AQ, BR, CW and IK: conceptualisation of the study. Data processing: AQ, with assistance from BR and CW. Formal analysis: AQ with interactions from all co-authors. Preparation of the manuscript: All co-authors reviewed the manuscript and contributed to the writing and final editing



*Competing interests.*

At least one of the (co-)authors is a member of the editorial board of Ocean Science.

*Acknowledgements.* This research has been supported by the international Multidisciplinary drifting Observatory for the Study of the Arc-
tic Climate (MOSAiC) with the tag MOSAiC20192020 (grant nos. AWI_PS122_00 and AFMOSAiC-1_00); the Alfred-Wegener-Institut
Helmholtz-Zentrum für Polar- und Meeresforschung (Bremerhaven, Germany) through the Multidisciplinary Icebased Drifting Observatory
(MIDO) infrastructure, the project AWI_OCEAN, and the project Sub-Mesoscale Dynamics and Nutrients (SMEDYN) within the INterna-
tional Science Program for Integrative Research in Earth Systems (INSPIRES); the EPICA project under the research theme MARE:N –
Polarforschung/MOSAiC, funded by the German Federal Ministry for Education and Research (grant no. 03F0889A); the European Com-
mission (EU H2020 grant no. 101003472, project Arctic PASSION); and the AROMA (Arctic Ocean mixing processes and vertical fluxes of
energy and matter) project funded by the Research Council of Norway (grant no. 294396). Language editing assistance was provided using
ChatGPT (OpenAI).





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
