# Peer review of "Intrahalocline eddies in the Amundsen Basin observed in the distributed network from the MOSAiC expedition"

_EGUsphere, 2025_

## Author Comment (AC2)

**Answers to the reviewer's comments**

December 8, 2025

We would like to sincerely thank the two reviewers for their supportive and constructive comments and suggestions, which greatly improved the manuscript. Our answers to the reviewer's comments are given below, in bold font.

On behalf of all co-authors,
Alejandra Quintanilla Zurita

**Reviewer 1**

Paper is on a topic of interest - detection and characterisation of subsurface eddies in the Arctic Ocean. These eddies potentially play a key role in mixing and so are of general interest to Arctic oceanography. Overall, I find the paper easy to follow, with a rigorous analysis of the data used in picking out of the eddies. Accordingly, I think the paper is a valuable contribution and will be of general interest.

However, at this stage the paper reads more like a data report than a scientific paper. For example, in the intro there is a detailed review of Arctic eddies, but there is virtually no discussion of what the eddies "do" within the context of the Arctic ocean system. Such comments would greatly strengthen the justification for the paper by linking to broader Arctic Ocean issues. This could easily be added by referring to recent reviews of the topic (eg. von Appen et al, 2022 - which is already referred to a different context, and Lenn et al (2021). Ocean Mixing: Drivers, Mechanisms, Impacts. Meredith, M. & Garabato, A. (eds.). Elsevier, p. 275-299).

With regard to the discussion on generation mechanisms I believe this discussion could also be usefully widened in an Arctic context. (eg. see MacKinnon et al (2021). A warm Jet in a cold ocean. Nature Comms, 12, 1, p. 2418; Schultz et al (2021). Turbulent mixing and the formation of an intermediate nepheloid layer above the Siberian continental shelf break. Geophys Res Letts).

**We thank the reviewer for this helpful suggestion. We have now expanded the Introduction to explicitly discuss the role of intrahalocline and subsurface eddies in the Arctic Ocean system, including their influence on halocline maintenance, upper-ocean heat content, vertical heat fluxes toward the sea ice, and lateral exchange between**

boundary currents and the basin interior. As recommended, we now cite von Appen et al. (2022) and Lenn et al. (2021) in this context. These additions clarify the broader significance of Arctic eddies and strengthen the motivation for the study. Changes made in the manuscript: Introduction, paragraph 5 (lines 60-70).

We agree and have incorporated a broader Arctic context in the section on eddy generation mechanisms. We now discuss how similar processes—such as instabilities induced by topographic jets, shelf–basin density fronts, and mixing associated with lead refreezing—have been documented across the Arctic Ocean. Following the reviewer's suggestions, we now explicitly cite MacKinnon et al. (2021) and Schulz et al. (2021), integrating these examples to frame our interpretation of the observed eddies and to highlight parallels with mechanisms proposed in other Arctic regions. Changes made in the manuscript: Section 4.2 "Origin and generation of eddies" (lines 378–390).

**Reviewer 2**

1) Include figures similar to Figures 4 and 6 for all detected eddies as supplementary material. This would provide a complete visual reference for each identified IHE and enhance the reproducibility and transparency of the analysis.
**We include figures equivalent to Figures 4 and 6 for all detected IHEs in Appendix A (Figures A1–A5).**

2) The statement that eddy generation and trajectories cannot be directly observed under Arctic sea ice is repeated several times throughout the manuscript. While this is an important contextual limitation, mentioning it once—ideally in the Introduction or early in the Discussion—would be sufficient.
**We now mention this limitation only once, in the introductory section of the Discussion, and have removed the repeated statements throughout the manuscript.**

- L6-7: Consider removing the explanation of the ADCP acronym, as it is a widely known instrument within the physical oceanography community.
**We retain the explanation of the ADCP acronym because this manuscript targets a broad interdisciplinary audience, and defining key terminology ensures clarity and accessibility for all readers.**

- L12-14: ".. ., placing them in the (sub)mesoscale dynamical regime, consistent with quasi-geostrophic behaviour.". Later sections report Rossby numbers up to 0.62, implying partial cyclogeostrophic balance. Suggest rewording to reflect a transitional dynamical regime where both geostrophic and cyclogeostrophic effects may be relevant.
**We revised the sentence to explicitly state that the observed eddies fall within a transitional dynamical regime in which both geostrophic**

and cyclogeostrophic effects can be relevant. "The observed eddy scales lie close to or slightly below the first baroclinic Rossby deformation radius of approximately 6.9 km, placing them in the (sub)mesoscale dynamical regime and suggesting a transitional balance where both geostrophic and cyclogeostrophic effects may be relevant."

- L18: "...suggest that local convection..." Replace with "shallow local convection" for clarity and to avoid implying deep convection processes. **Done.**

- L34: "...which may exhibit diameters close to or below this threshold." The use of "this threshold" is confusing, as the Rossby radius has not been explicitly introduced as a limiting value. Suggest replacing with "this scale" or "this length scale" for clarity. **Done.**

- L38: "The Amundsen Basin is the deepest part of the Arctic Ocean...". Consider referencing Fig. 1a here, since this figure displays the bathymetry and general setting described in lines 38–44. **Done.**

- L46: "...where horizontal velocities are minimal at the eddy core..." This statement is not strictly accurate as written. Velocities are minimal at the centre of the core, not uniformly "at the core." Please clarify this radial structure. **We revised this sentence to more accurately describe the radial velocity structure of the eddy: "In general, eddies are characterised by a maximum vertical displacement of isopycnals at their centre, consistent with geostrophic balance, and horizontal velocities reach a minimum at the eddy centre and increase radially outward within the solid-body core (Zhao and Timmermans, 2015)".**

- L47–49: The manuscript presents the contrasting vertical isopycnal structures of anticyclonic vs. cyclonic IHEs and cites McGillicuddy Jr. (2015). Is there observational support for cyclonic IHEs in the Amundsen / Eurasian Basin, or elsewhere in the Arctic, beyond this reference? If McGillicuddy Jr. (2015) is being used conceptually rather than as direct observational evidence under sea ice, consider softening the phrasing. **We clarified that the contrasting vertical isopycnal structures of anticyclonic and cyclonic eddies follow theoretical expectations and previous observations from other Arctic regions. We also state explicitly that no cyclonic intrahalocline eddies were identified in our dataset. "Anticyclonic IHEs typically exhibit domed isopycnals above and depressed isopycnals below. The opposite vertical structure is theoretically expected for cyclonic subsurface eddies: depressed isopycnals above and domed isopycnals below (e.g., McGillicuddy Jr, 2015; Zhao et al., 2014), although such features have not been documented as**

**IHEs in the Amundsen Basin and were not observed in our dataset."**

- L50: The discussion of prior studies on IHEs/ITEs would benefit from one or two additional key references, both theoretical and observational, to frame IHEs in the broader context of intrathermocline eddies and subsurface lenses. I suggest including reference such as: McWilliams (1985, RGO, doi: 10.1029/RG023i002p00165; 1988, JPO, doi: 10.1175/1520-0485(1988)018¡1178: VGTBA¿2.0.CO;2); Chaigneau et al. (2011, JGR, doi: 10.1029/2011JC007134); Dilmahamod et al. (2018, JGR, doi: 10.1029/2018JC013828); among others. These help connect Arctic IHEs to the global literature on sub-surface, density-trapped vortices.

**We expanded the background section to place Arctic IHEs within the broader global literature on subsurface, density-trapped vortices, incorporating the suggested theoretical and observational references: "These features belong to the global class of density-trapped subsurface vortices commonly termed intrathermocline eddies or subsurface lenses (e.g., McWilliams, 1985, 1988; Chaigneau et al., 2011; Dilmahamod et al., 2018). In the Arctic Ocean, where stratification is predominantly halocline-controlled, similar features have been described as IHEs (e.g., Zhao et al., 2018; Fine et al., 2018). We therefore adopt this terminology."**

- L50–57: The paragraph lists previous studies (Aagaard and Carmack, 1989; Manley and Hunkins, 1985; Timmermans et al. 2008; Zhao et al. 2014; Polyakov et al. 2012; Woodgate et al. 2001) but does not synthesize their key dynamical/structural findings. I strongly recommend adding 1–2 summary sentences describing what those studies actually found in terms of typical radii, vertical extent / core depth, maximum azimuthal velocity, water mass properties (e.g. Canadian vs. Eurasian halocline waters), Rossby number / dynamical regime if available. This would give the reader a quantitative baseline of "what an Arctic IHE usually looks like," against which the present MOSAiC eddies can be evaluated as similar, distinct, or extreme.

**We expanded the paragraph to synthesise the structural and dynamical characteristics reported by previous studies, providing a quantitative baseline for comparison with the MOSAiC eddies: "Previous studies of such eddies, mainly in the Canada Basin (e.g., Aagaard and Carmack, 1989; Manley and Hunkins, 1985; Timmermans et al., 2008; Zhao et al., 2014), contrast with sparse evidence from the Eurasian Basin, mostly from limited mooring observations (Polyakov et al., 2012; Woodgate et al., 2001). Zhao et al. (2014) identified 39 eddies with Eurasian Basin water from a decade of Ice-Tethered Profiler (ITP) data, but most were found on the Canadian side, rarely in the Amundsen Basin, confirming that the region remains poorly sampled and characterised. Their study showed that these Arctic subsurface eddies are predominantly anticyclonic, with radii of 3.5-7 km, core depths of 54–150 m, mean azimuthal velocities of 0.05–0.22 $ms^{-1}$, and**

Rossby numbers of 0.07–0.63. Beyond their structure, these eddies redistribute water and heat within the halocline, affecting halocline maintenance, mixed layer properties, and upper-ocean heat content in a changing Arctic (e.g., Von Appen et al., 2022; Lenn et al., 2022)".

- Figure 1a: The bathymetric colour scale should be inverted, this would align with conventional bathymetric visualisation practices and improve intuitive interpretation; add labels for the main bathymetric and geographic features visible in the panel, such as Lomonosov Ridge, Gakkel Ridge, Fram Strait, and the East Siberian Shelf; include a simple schematic arrow indicating the Transpolar Drift, Artic Ocean Boundary Current, and the entrance of Atlantic Water Boundary Current.

**We implemented all suggested improvements. The bathymetric colour scale is now inverted following standard conventions, and major bathymetric and geographic features (Lomonosov Ridge, Gakkel Ridge, Fram Strait, East Siberian Shelf) have been added. We also included schematic arrows indicating the Warm Atlantic Water Boundary Current and the Transpolar Drift.**

- Figure 1b: Here, the spatial configuration of the L-sites around the CO on 10 October 2019 is shown. It is unclear why this specific date was chosen, since the study period spans 19 October 2019 to 15 March 2020. Was this the configuration at deployment, or does it represent the average geometry during the drift? Please clarify the rationale for selecting this date. If the configuration changed noticeably during the winter drift, consider showing the mean position with a range of variation (e.g., using error bars, semi-transparent ellipses, dashed outlines, or a numerical range such as mean $\pm$ interval) to illustrate the spatial evolution of the DN. Relatedly, the DN likely rotated or sheared around its central axis during the drift. Quantifying this rotational or deformation rate (if available) would help readers evaluate whether such motion could affect the quasi-synoptic assumption or the relative sampling geometry used in the eddy detection analysis. As a visual aid, you could optionally include a transparent ellipse indicating the approximate range of motion or orientation of the L-sites around the CO, aligned with the mean propagation axis. This would communicate both spatial uncertainty and the general drift direction in a single glance. Rabe et al. (2024) includes snapshots and maps showing DN behaviour and evolution during MOSAiC. If these aspects are already documented there, please explicitly refer to that study in the main text when describing the DN configuration adding its temporal variability.

**We updated Figure 1b to use the same starting date as the drift track (19 October 2019). In addition, we now show the relative position of each L-site with respect to the CO–PS using the same colour scale as in panel (a), ensuring visual consistency across the figure.**

- Figure 1c: Indicate on the time series the specific moments when the IHEs were detected (e.g., using coloured markers or shaded bands). This would help

the reader relate the eddy occurrences to variations in drift and current speed. Consider adding a new complementary panel (e.g., Fig. 1d) showing the "mean vertical profile of current magnitude", one including all and others separated into periods when the instruments were inside IHEs and when they were outside IHEs. This comparison would visually demonstrate the influence and characteristic velocity enhancement or structure associated with eddy passages. The caption currently refers to "mean current speed averaged from the mixed layer to 100 m," but not all ADCP records reached 100 m depth. Please revise this depth range to reflect the actual valid measurement interval (e.g., "averaged over the available ADCP depth range" or specify the precise depth range used). **We updated panel (c) by marking each eddy (E1–E9) on the top axis, allowing direct comparison between drift speed, current speed, and eddy passages. We also revised the averaging depth range to reflect the actual ADCP coverage at each site. "Drift speed of the CO (grey line) and the mean current speed averaged over the available ADCP depth range at CO-PS (green line), L1 (blue line), and L3 (red line). The dashed grey and black lines show the mean drift and mean current speed, respectively. The timing of each detected eddy (E1–E9) is indicated at the top of the panel".**

[Figure]

Figure 1: (a) Drift track of the Central Observatory (CO) on the MOSAiC expedition from October 19 2019 to March 15 2020. The blue scale represents the bathymetry ($\times 10^3$ m), based on the International Bathymetric Chart of the Arctic Ocean (Jakobsson et al., 2008). Schematic arrows indicate the major upper-ocean circulation pathways: the Warm Atlantic Water Boundary Current (pink) and the Transpolar Drift (red). Key bathymetric and geographic features are labelled, including the Lomonosov Ridge (LR), Gakkel Ridge (Gk), Fram Strait (FS), and the East Siberian Shelf (ESS). The black polygon in panel a shows the main area of study. (b) Initial spatial configuration of the L-sites relative to the CO–PS on 19 October 2019 (markers), and their subsequent relative displacements from 19 October 2019 to 15 March 2020 (colour-coded positions). The colour scale matches that of panel (a), indicating the date along the drift trajectory. (c) Drift speed of the CO (grey line) and the mean current speed averaged over the available ADCP depth range at CO-PS (green line), L1 (blue line), and L3 (red line). The dashed grey and black lines show the mean drift and mean current speed, respectively. The timing of each detected eddy (E1–E9) is indicated at the top of the panel.

- General for "Data" subsection (end of first paragraph, L70–84): The paragraph effectively introduces the dataset and deployment configuration, but

it ends abruptly without describing how the data were processed. It would strengthen transparency and reproducibility to **include a brief description of the processing workflow** — either as a step-by-step summary (e.g., sampling frequency, sensor type, software used, averaging, interpolation, filtering, de-tiding, etc.) or by citing an established data-processing scheme from previous works (e.g., Krishfield et al., 2008, JTECHO, doi: 10.1175/2008JTECHO587.1; Timmermans et al., 2010, JTECHO, doi: 10.1175/2010JTECHO772.1; Toole et. 2011) This would also clarify whether identical methods were applied to all ITPs and ADCPs or adapted for specific platforms.

**We now clarify that all observations were used as provided in their original, quality-controlled form from the respective MOSAiC data products. We added explicit citations to each dataset and specified that only a low-pass filter was applied to the velocity records to remove high-frequency variability. "All datasets were used in their publicly released, quality-controlled form from the official MOSAiC data products, including the ITP (Toole et al., 2016), the Polarstern sADCP (Tippenhauer and Rex, 2020), the AOFB (Stanton and Shaw, 2023), and the Polarstern CTD (Tippenhauer et al., 2023). No additional corrections, averaging, or interpolation were applied. Only the velocity data were smoothed with a half-day low-pass filter to reduce high-frequency noise while retaining eddy-scale variability. This was done to remove the short-period noise while preserving temporal variability at time scales expected for (sub)mesoscale eddies, which typically last several days".**

- L85–88 and Table 1: The description of the ITP and ADCP sampling configurations in lines 85–88 is difficult to follow, and the same information in Table 1 is not immediately clear without reading the accompanying text. The table header should be improved to make it self-contained. Consider revising the table caption and column headings to explicitly state what each variable represents (e.g., "Profiling frequency [h]" = interval between consecutive profiles, "Vertical resolution [m]" = bin size of the instrument). Include units directly in the header rows and, if possible, a short note indicating the meaning of abbreviations such as AOFB, ITP, and CO-PS. The text in lines 85–88 could also be simplified to describe, in plain terms, that each site had different profiling intervals and vertical ranges, which can be cross-checked directly in the table. This would make Table 1 more intuitive and reduce the need for readers to cross-reference multiple parts of the manuscript to interpret the experimental setup.

**We agreed that this section was unclear; we therefore implemented the suggested improvements by revising the table caption to explicitly define all variables and units (Location site, Platform (buoy system), Sensor type, Deployment period, Time between profiles [h], Depth range [m], Depth bin size [m]). We also merged the two descriptive paragraphs, directed readers to Table 1 for instrument-specific details, and briefly clarified the reasons for missing or incomplete**

records. "The three L-sites were instrumented with ITPs and AOFB-mounted ADCPs, each operating with different profiling intervals and vertical sampling ranges (Table 1). Because all platforms drifted with the sea ice, the horizontal spacing between consecutive profiles depended on both drift speed and profiling interval, ranging approximately from 1 to 10 km, with the L3 ITP providing the smallest spacing and L2 the largest. The L1 and L2 sensors remained operational throughout the winter and drifted toward the Fram Strait, whereas the L3 sensors ceased operation on 31 January 2020 following an ice-ridging event. The L2 ADCP did not return usable data due to early technical failure, and when L2 and the Central Observatory were aligned, velocity measurements were supplemented using the Polarstern shipboard ADCP."

- L88–89: The sentence "Given the drift velocity of the sea ice,..., where L3 has the highest horizontal resolution and L2 the lowest" is somewhat confusing as written. The expression "horizontal resolution" may be misleading here, since the authors refer to the horizontal spacing between consecutive profiles, which results from the product of the ice-drift speed and the profiling interval. I suggest rephrasing for clarity, for example: "Because the ITPs drifted with the sea ice, the horizontal separation between consecutive profiles depended on the drift speed and profiling frequency, ranging from about 1 km (at L3) to 10 km (at L2)". In addition, since the drift velocities and profiling intervals of the L-sites are known, it should be possible to compute and visualise the variability of the network geometry throughout the experiment. As mentioned above, these data could be used to support Figure 1b, quantifying how the spatial configuration of the DN evolved over time.

**We rephrased this sentence to clarify that the spacing between consecutive profiles results from the combination of sea ice drift speed and the profiling interval. "Because all platforms drifted with the sea ice, the horizontal spacing between consecutive profiles depended on both drift speed and profiling interval, ranging approximately from 1 to 10 km, with the L3 ITP providing the smallest spacing and L2 the largest." As written above, we also modified Figure 1b to visualize the variability of the network geometry.**

- L90: Replace "ridging" with "ice-ridging" for clarity.
**Done.**

- L92–95: The description of the ADCP configurations and measurement ranges in these lines does not fully match the information presented in Table 1 (e.g., time coverage, vertical range, and resolution). Please revise to ensure complete consistency between the main text and the table.
**We revised this section to ensure full consistency with Table 1. To avoid redundancy and potential confusion, we now retain all ADCP configuration details exclusively in the Table 1.**

- L97–103: The assumption of quasi-synoptic conditions is reasonable, yet it would be useful to discuss possible biases introduced by the relative motion between the drifting DN array and the eddies. Specifically, the direction of eddy propagation, the spatial separation between L-sites, and any rotation or deformation of the DN configuration could influence how individual profiles sample the evolving eddy field. These effects might alter the apparent azimuthal velocity structure or introduce Doppler-like distortions in the reconstructed sections. I recommend acknowledging these potential limitations and, if possible, estimating their magnitude (see Allen et al., 2001, Deep-Sea Research, doi: 10.1016/S0967-0637(00)00035-2) or citing supporting work that discusses how sampling geometry and motion relative to a propagating eddy can affect quasi-synoptic interpretations.

**We thank the reviewer for the thoughtful comment regarding potential geometric biases associated with the relative motion between the drifting DN array and propagating eddies. We agree that such effects can influence quasi-synoptic assumptions in multi-platform or gridded sampling configurations (e.g., Allen et al., 2001). In our observational setting, however, these distortions are expected to be minimal for three reasons: First, the DN geometry remained stable throughout the drift (Fig. 1b), with only minor changes in inter-platform distances and no significant rotation or deformation of the array. The separation between L-sites ($\geq$ 9 km) was always larger than the expected diameter of Arctic intrahalocline eddies ($\approx$ 2–5 km), making it physically unlikely for a single eddy to be sampled simultaneously or sequentially by multiple platforms. Thus, the multi-platform sampling distortions described in Allen et al. (2001) do not apply here. Second, the sea ice drift advected the instruments at a mean speed of 0.1 $ms^{-1}$, whereas subsurface eddy propagation speeds were an order of magnitude slower ($\approx$ 0.01–0.02 m $ms^{-1}$, Fig. 1c). This large velocity difference ensures that the platforms move faster than the evolving eddies, substantially reducing any Doppler-type distortions and supporting the quasi-synoptic "frozen-field" assumption commonly used in ice-tethered observations. Third, to further minimise geometric biases, we restricted our eddy identification to segments in which the platform trajectories were approximately linear during the eddy encounter. We added the following sentences in the manuscript: "In this study, the DN moved with the sea ice at a mean drift speed of 0.11 $ms^{-1}$, while the underlying ocean current below the mixed layer had an average speed of 0.02 $ms^{-1}$ (Figure 1c). Because ice drift is an order of magnitude faster than ocean currents, the ice-tethered platforms move quickly relative to the ocean features beneath them. The DN geometry also remained stable during the drift (Figure ??b), with inter-platform distances changing only slightly and always exceeding the expected diameter of Arctic intrahalocline eddies. This large difference in speeds and the absence of significant deformation**

or rotation of the array justifies the quasi-synoptic assumption, which means that measurements from the ice-advected platforms can be considered as near-instantaneous snapshots ("frozen fields") of the slower-evolving ocean eddies (Manley and Hunkins, 1985; Krishfield et al., 2008). Furthermore, the analysis was restricted to DN trajectories that followed approximately linear paths during each eddy encounter, thereby minimising potential geometric biases arising from the relative motion between the platform and a propagating eddy. This interpretation agrees with observations that Arctic eddies propagate at speeds roughly an order of magnitude slower than the sea ice drift (von Appen et al., 2018)".

- L104–113: The description of how isopycnal displacements were used to identify eddies is somewhat unclear. From the context, the method appears to rely on the temporal evolution of vertical density structure, interpreted as spatial variability because the DN drifted fast enough to cross an IHE during successive profiles. This implies that the identification is primarily visual, based on isopycnal tilting patterns assumed to correspond to eddy structures. The explanation should focus explicitly on how IHEs were recognised in the ITP data. If all detected features are anticyclonic, this should be clearly stated. Additionally, note that in IHEs, the isopycnal slopes are opposite at the upper and lower limits of the eddy core. Making this explicit would help readers visualise the identification criteria more accurately.

**We improved the explanation of the isopycnal displacement method, clarifying how eddies were identified from consecutive ITP profiles, the expected vertical structure of anticyclonic IHEs, and noting explicitly that all detected features were anticyclonic: "To identify eddies, we follow the methodology suggested by Timmermans et al. (2008) and Zhao et al. (2014). Eddies were first recognised in the ITP profiles by visually detecting coherent vertical displacements of isopycnals across several consecutive casts, reflecting the eddy's spatial structure as the drifting platform crosses it. In the case of the anticyclonic IHEs, the only type detected in our dataset, the upper part of the eddy shows a convex upward doming of isopycnals, whereas the lower part exhibits a concave downward displacement, producing opposite slopes above and below the core."**

- L107–109: It is unclear whether the criterion requiring "at least two profiles within the speed anomaly" refers to two profiles on each side of the maximum isopycnal displacement or one profile on each side. Please clarify this point explicitly.

**We clarified this criterion in the revised text. The method now specifies that the ADCP velocity anomaly must exhibit two local maxima in horizontal speed, one on each side of the isopycnal displacement centre, consistent with the expected azimuthal structure of an eddy.**

- L110: Please clarify what this "additional" refers to.

**We clarified this point by replacing "additional" with "combined ITP–ADCP observations," emphasising that the co-located velocity measurements strengthen the eddy identification in cases where only two consecutive profiles exhibit isopycnal displacement.**

- L114:Replace "defines the radius of maximum velocity" with "is defined by the radius of maximum velocity" to better reflect the physical relationship being described.

**Done.**

- L117: When mentioning "other features such as meanders or fronts," please include relevant references documenting their presence in the region. For example, studies by von Appen et al. (2018; 2022) describe submesoscale filaments, meanders, and frontal activity in the marginal-ice and central Arctic.

**We included the suggested references documenting submesoscale meanders and frontal intrusions in the central and marginal Arctic Ocean. "The last step in confirming that an eddy-like structure is in fact a rotating eddy is to verify that the velocity field is dynamically consistent with coherent rotation rather than with other features such as meanders or frontal intrusions, which have been documented in the central and marginal Arctic Ocean (e.g. von Appen et al., 2018; Von Appen et al., 2022; Zhao et al., 2014; Timmermans et al., 2008; Polyakov et al., 2012)."**

- L117–124: The methodology implicitly assumes that each L-site transected the IHE, yet it is not explained how this crossing was verified. There seems to be a missing step in the identification procedure: before testing for solid-body rotation, one should check for a change in the sign of the azimuthal (or cross-track) velocity, indicating opposite flow directions on either side of the eddy centre. This reversal is a key diagnostic confirming that the profiler indeed crossed through (or near) the eddy core rather than only sampling one flank. Please clarify whether such a criterion was applied.

**We clarified that a sign reversal in the azimuthal velocity is an explicit and necessary criterion before assessing solid-body rotation: "As a first diagnostic, we require that the azimuthal (cross-track) velocity $v_\theta$ exhibit a reversal in sign across the centre of the isopycnal displacement, indicating opposite flow directions on the two flanks of the feature and ensuring that the profiler crossed through or very near the eddy core. As a second diagnostic, we test whether $v_\theta$ increases approximately linearly with radius within the core, consistent with the solid-body rotation expected in mesoscale eddies (Nurser and Bacon, 2014); the radius of maximum velocity then marks the edge of the core (Chelton et al., 2011)".**

- L128: The citation appears to be misattributed. The approach referenced

here corresponds to Castelão et al. (2013), whereas Castelão & Johns (2011) focused on the North Brazil Current rings that informed this method's development.

**We acknowledge the clarification regarding citation provenance. The analytical form of the azimuthal velocity used in our fitting procedure was initially introduced in Castelão & Johns (2011), while Castelão et al. (2013) formalised its methodological application for velocity-profile fitting. We clarified this distinction in the revised manuscript and now explicitly reference both contributions.**

- L135: The text refers to "Equation 1" when describing the computation of azimuthal velocity, but based on the context, it should correspond to Equation (2). Please verify and correct the equation reference to maintain consistency with the definitions given earlier.
**Done.**

- L135–137: A short transition sentence is needed to connect this paragraph with the following subsection. As written, the discussion around the Rankine vortex model appears self-contained, but Figure 3 shows results refined through the Maximum Swirl Velocity (MSV) method explained in the next subsection. Without this clarification, readers may interpret the point of maximum velocity as being chosen arbitrarily for the Rankine fit. **We clarified in both the text and the figure caption that the Rankine comparison in Figure 3 uses the maximum azimuthal velocity directly measured from the ADCP data, computed using the methods introduced in this subsection. This avoids any ambiguity regarding the choice of the scaling velocity and makes clear that no additional refinement (introduced later) affects the results shown in Figure 3.**

- L147: It is unclear how is obtained at this stage.
**We clarified how the search radius is obtained at this stage by explicitly describing the gridded search procedure introduced by Nencioli et al. (2008). The revised text now explains that we construct a $2R_{max} \times 2R_{max}$ search domain around the location of minimum measured velocity, discretise it at 100 m resolution, and evaluate the MSV method at each candidate centre. This makes clear how the search region is defined and how the centre estimate is refined before computing the cost function $J$.**

- L149: Change "Equation 1" to "Equations (1–3)", as all three are used in the velocity decomposition.
**Done.**

- L151: Specify which optimisation method was used to minimise the cost function J.
**We clarified the optimisation procedure by specifying that the cost**

function $J$ was evaluated at every point of the gridded search area, and the eddy centre was identified as the grid point where $J$ reached its minimum.

- L155: The variable Rmax mentioned here appears to correspond to the refined radius obtained from the MSV method, which is later denoted as Rm in the text and in Table 2. Please standardise the notation throughout the manuscript to avoid confusion.
**Done.**

- L169: It seems that the reported Coriolis parameter contains both a unit error and a missing exponent in the numerical value.
**Correct value is changed to $f = 1.45 \times 10^{-4} s^{-1}$.**

- L172–173: The statement "This area is distant from major bathymetric features..." is not entirely accurate, since the Amundsen Basin itself is bounded by prominent topographic features such as the Lomonosov and Gakkel Ridges. Please clarify how the study region can nonetheless be approximated as laterally homogeneous and vertically unaffected by topography. Explaining this assumption would help justify that, given the small Rossby deformation radii in this area, such an approximation remains appropriate for the present analysis.
**We clarify this by changing the text: "Although the Amundsen Basin is limited by the Lomonosov and Gakkel Ridges, our study area lies in the interior of the basin, where the upper-ocean stratification and water-mass structure are known to be horizontally uniform and largely independent of ridge-controlled dynamics (Rudels et al., 1996)".**

- Figure 2: Include position of CTD and ADCP profiles as in Fig. 6
**Done.**

- L184–185: Please clarify the distinction between the solid-body rotation region (core) and the inner core.
**We thank the reviewer for noting this ambiguity. In the revised manuscript, we avoid introducing unnecessary terminology by referring only to the eddy core, defined as the region of solid-body rotation. The term "inner core" has been removed to maintain consistency and prevent confusion.**

- L187: The phrase "best-sampled eddies" is vague. Specify the criterion used. Including in Table 2 the number of CTD and ADCP profiles used for each eddy (or within each eddy's core) would make this classification transparent.
**We change the phrase "best-sampled eddies" with a more precise description: "they were the largest and most energetic detected along the drift track, with strong azimuthal velocities and clear hydro-**

**graphic signatures representative of wintertime IHEs". Also, we add the number of ITP and ADCP profiles for each eddy in Table 2**.

- L197–200: It is not clear how the mixed-layer depth (MLD) was determined. Please clarify the method in the Data subsection, alongside the data-processing description and the use of TEOS-10 for thermodynamic variable computation.

**We add on data section: "The MLD was defined as the first depth at which the Brunt–Vaisala frequency anomaly, $\Delta N$, between successive measurements exceeded $3 \times 10^{-4} s^{-2}$. This threshold was selected following a visual inspection of all available profiles, as it reliably captured the transition from the mixed layer to the onset of the halocline during the MOSAiC drift. All thermodynamic variables, including density, $N$, and derived quantities, were computed using TEOS-10 through the GSW Python toolbox (McDougall and Barker, 2011)".**

- L200–201: The description is somewhat confusing, as the vertical boundaries of the IHEs were defined using local maxima in buoyancy frequency (N). Are these N maxima associated with specific isopycnal surfaces? If so, please indicate whether the corresponding isopycnal densities were consistent among the observed IHEs or varied from case to case.

**We have now specified that the local maxima in buoyancy frequency are used to identify the isopycnal surfaces that bound each eddy vertically, and we explicitly state how these are selected. "The $N$ profiles were used to identify the isopycnal surfaces that bound each eddy vertically. We define the upper limit of the eddy as the isopycnal coincident with the depth of the first peak in $N$, and the lower limit as the isopycnal coincident with the depth of the second peak in $N$".**

- Figure 6: Please include absolute salinity (SA) sections and profiles in this figure to complement the conservative temperature and density fields. If adding them here would overcrowd the panel, consider including the SA plots as supplementary material for completeness. Including this data is essential to show how each eddy is embedded in the vertical salinity structure and to visualise the associated halocline displacement.

**We agree with the suggestion and have revised Figure 6 accordingly. To avoid overcrowding the panel, we split the original Figure into two, allowing us to include the Absolute Salinity (SA) sections and vertical profiles.**

[Figure]

Figure 6: Details of the eddies E8 (upper panel) and E9 (lower panel). (a) Cross sections of conservative temperature ($\Theta$) and (b) absolute salinity ($g\ kg^{-1}$) with isopycnals shown as black contours spaced every 0.25 $kg\ m^{-3}$; red and cyan contours indicate the upper and lower limits of the eddies, respectively, and the dashed vertical red line marks the central eddy profile. (f) Cross section of azimuthal velocity $v_\theta$ with velocity contours in grey every 0.05 $ms^{-1}$. The green dotted line indicates the depth of maximum velocity. The dotted vertical light grey lines in (a), (b) and (f) marker the measurement profiles, where the darker lines are the profiles inside the eddy in Figure 7.

[Figure]

Figure 7: Details of the eddies E8 (upper panel) and E9 (lower panel). Vertical profiles of density ($\sigma_\theta$) (a), absolute salinity ($g\ kg^{-1}$) (b), buoyancy frequency (N) (c), conservative temperature ($\Theta$)(d), and azimuthal velocity ($v_\theta$) (e). The red line shows the central profile, and the grey line shows the mean profiles at $\pm 20$ km around the eddy, marked as dotted vertical light grey lines in Figure 6. Dashed horizontal lines show the top (red), the bottom (cyan), the maximum azimuthal velocity level (green) and the eddy core-centre depth (orange).

- L221: The refined radii obtained through the MSV method are denoted as Rm. Please ensure this notation is used consistently throughout the text and figures, replacing any remaining instances of where appropriate.
**Done.**

- L225–226: Consider whether geostrophic currents could be estimated from the thermal-wind balance, at least for the well-resolved IHEs at the L3 site. This would allow comparison with directly measured velocities and provide insight into the relative contribution of horizontal ageostrophic velocity components. It would also be valuable to discuss this in the context of cyclogeostrophic balance, possibly citing Shakespeare (2016, JPO, doi:10.1175/JPO-D-16-0137.1)

to strengthen the argument regarding the transitional regime described later. Additionally, consider computing or at least discussing the length-scale Burger number, as this parameter could clarify the dynamical interpretation presented in lines 235–240 and Figure 8.

**We appreciate the reviewer's suggestion and initially attempted to estimate geostrophic and cyclogeostrophic velocities from the ITP profiles. However, this was not feasible for our dataset. As noted by Zhao (2014), dynamic height calculations based on ITP data must rely only on upcasts, as downcasts are affected by wake-induced thermal and conductivity biases. This constraint reduces the number of usable profiles by half. Although the vertical resolution of each profile is high (1 m) and thus adequate for resolving vertical shear, the main limitation is horizontal sampling: for most eddies, we only have a few ITP profiles within the eddy radius, typically not more than two profiles on each side of the core. Such sparse sampling does not provide a robust estimate of the cross-eddy dynamic-height gradient required for thermal-wind or cyclogeostrophic calculations, especially for eddies with radii of only a few kilometres. Only one eddy (E8) marginally satisfied the sampling requirements, but we opted not to pursue a single-eddy analysis since the goal of the study is to characterise the full ensemble. While direct geostrophic estimates were therefore not feasible, we expanded the dynamical interpretation in two ways. First, we now explicitly discuss the relevance of cyclogeostrophic balance in the transitional Rossby-number regime observed in our eddies, citing Shakespeare (2016). Second, we computed the Burger number for each eddy ($Bu = (R/L_1)^2$), which ranges from 0.4 to 2.9. These values indicate that most eddies lie in a submesoscale-to-mesoscale transitional regime ($Bu \approx 1$), while a minority with $Bu > 1$ are more compact and consistent with partially cyclogeostrophic structures. These additions clarify the dynamical context of our eddy interpretation despite the lack of direct geostrophic velocity estimates. "the Burger number ($Bu$) computed for the nine eddies (0.4–2.9) further show that most lie in a submesoscale-to-mesoscale transitional regime ($Bu \approx 1$), while a minority ($Bu > 1$) exhibit more compact structures where cyclogeostrophic effects may become relevant. We computed $R_o$ using the cylindrical approximation $Ro = \frac{2U}{fR}$ (Zhao, 2014), where $U$ is $V_{max}$, $f$ is the Coriolis parameter and $R$ is the radius. Similarly, the Burger number was computed as $Bu = (\frac{R}{L_1})^2$, where $L_1$ is the first baroclinic Rossby deformation radius. This yields $R_o$ (0.16 ¡ $R_o$ ¡ 0.62) that is consistent with the quasi-geostrophic balance, although the upper range allows for curvature effects (e.g., Shakespeare, 2016)."**

- L230-231, 237, Figure 7 and 8: Please define explicitly how the hydrographic "core properties" were calculated.

**We now clarify explicitly how hydrographic core properties were com-**

puted. Throughout the revised manuscript, "core properties" refer to the hydrographic values at the core-centre, defined as the depth where the buoyancy frequency $N$ reaches its minimum between the upper and lower isopycnal boundaries of the eddy. This distinction makes clear that the core denotes the full volume bounded by these two isopycnals and the radius of maximum azimuthal velocity, whereas the core-centre corresponds to the point of minimum stratification within that volume, from which the reported properties are extracted.

- L238 and Figure 8c: The claimed linear relationship in Figure 8c is not clearly visible. However, this lack of correlation is still an interesting result, it could reflect the degree to which centripetal forces contribute to the eddy's dynamical balance or the limited resolution of some eddies due to fewer available profiles. You may consider highlighting the best-sampled eddies (e.g., with different symbols) and performing a linear fit only for those cases, or alternatively, revising the interpretation to acknowledge the absence of a clear linear trend.
We revised the interpretation of the $V_{max}-$ thickness relationship. Rather than describing it as linear, we now report the presence of two distinct groups: one of eddies with lower $V_{max}$ and $Ro \approx 0.2$ close to geostrophic balance, and another with higher $V_{max}$ and $Ro \approx 0.3$–0.6 that require stronger azimuthal velocities, indicating a transition to cyclogeostrophic balance (consistent with Zhao et al., 2014).

- L243: This idea could be expanded further. Consider performing a non-parametric statistical analysis (e.g., clustering or rank-based test) to assess whether the Rossby number (Ro) values support a natural segmentation of the eddy population. Such an analysis would help justify the chosen Ro thresholds shown in Figure 8b and strengthen the interpretation of dynamical regimes.
We appreciate the reviewer's suggestion. While we decided not to perform a formal non-parametric clustering analysis, we expanded the discussion to clarify that the Rossby number distribution itself exhibits a natural segmentation. This separation between a low Ro group ($Ro \approx 0.2$) and a higher Ro group ($Ro \approx 0.3 - 0.6$) is consistent with what an exploratory, model-free clustering approach would typically identify. We now explicitly note that this natural grouping supports our interpretation of two dynamical regimes: a predominantly geostrophic regime for the low Ro eddies and a cyclogeostrophic regime for the higher Ro eddies.

- L245: Correct the figure reference; the text refers to Figure 8d, not Figure 7d.
Done.

- Table 2: Add a brief definition of what "core values" represent. Also specify what stands for Rm in this table and include its units. I assume these correspond to the refined radii, but this should be stated explicitly.

**Done.**

- General (Section 4.1): It is not clear why the analysis of potential duplicate detections focuses only on L1, L3, and CO, while L2 is excluded. Since the text repeatedly notes that duplicate sampling is most likely between L2 and CO, it would be appropriate to explain why these observations were not analysed or illustrated. A brief justification (e.g., data gaps, unreliable velocity records, or spatial separation) would clarify the rationale for this choice and ensure consistency with the earlier statements in the manuscript.

**Although duplicate detections are most likely between L2 and the CO, the L2 site does not provide velocity measurements, preventing a full application of our duplicate detection criteria. Only in two cases (E7 and E9) did the drift geometry cause the CO-PS ADCP and the L2 ITP to consecutively sample the same location, allowing us to identify these as duplicates based on the CO velocity anomaly combined with the L2 hydrographic signal. These are the only eddies detected at L2, and no other L sites recorded eddies during these periods. We now explain this explicitly in the text to clarify why L2 is included only in these specific cases and not in the broader multi-site comparison. "In the specific cases of E7 and E9, the CO ADCP detected an eddy shortly before the L2 ITP sampled a similar signal. Because the DN drifted northeastward–southeastward, both platforms consecutively passed over the same region, allowing us to conclude that L2 and CO sampled the same eddy. These are the only two eddies detected at L2, and since L2 does not provide velocity measurements, such events represent the only situations in which L2 can be meaningfully compared with CO. Moreover, during the periods when E7 and E9 were observed, none of the other L sites (L1 or L3) recorded eddy signals, further limiting the usefulness of L2 for the multi-site duplicate-detection analysis presented below".**

- L256-258: Consider discussing potential rotational effects of the DN during the drift, and please also indicate the expected translational velocity of the IHEs.

**We now clarify that the Distributed Network (DN) experienced rotation during its drift. However, the relative geometry and distance remained approximately constant throughout the observation period (Fig. 1b). Such rotations did not substantially modify the distances between platforms, and therefore did not create configurations in which two sites could simultaneously or quasi-simultaneously sample the same eddy. We also added the expected translational velocity of intrahalocline eddies (IHEs), which we estimate to be on the order of $0.02\ ms^{-1}$, consistent with previous observations and with the background flow in the Amundsen Basin. This slow propagation further supports the assumption of quasi-synoptic sampling and reinforces the argument that a single eddy is unlikely to cross multiple sites**

within a single observation window. We modified the text as follows: "The mean background flow of the Transpolar Drift, about **0.02 $ms^{-1}$, advects the IHEs at approximately the same speed (Zhao et al., 2014), further limiting the distance an eddy can travel between consecutive profiles. Although the DN rotated during the drift, its overall configuration and relative distances remained effectively constant throughout the study period (Figure 1b). Therefore, rotation does not alter the spatial separation between sites nor create conditions under which a single eddy could be sampled simultaneously at different locations".**

- L260: Please include the range of variation for this distance.
**Done: "Given the spatial separation among the DN sites, L1–L3 ≈ 32–35 km, L1–CO ≈ 10–17 km, and L3–CO ≈ 22–24 km, it is therefore unlikely that the same eddy would be sampled at more than one location, except in the few cases where the drift geometry brought two platforms over the same region within a short time interval".**

- L261: Include the distance between L2 and the CO for completeness and consistency with the other site descriptions.
**Done: "... separated by approximately 9–14 km".**

- L269: The text incorrectly states that L3 detected IHE E4 on 29 October, whereas Figure 9 clearly shows that L3 detected E2, not E4. Please correct this reference. In addition, the description should explicitly refer to the corresponding panels in Figure 9 that illustrate these detections to help readers follow the discussion. **Done.**

- L270-271: This sentence should be rewritten for clarity. Visually, the two eddies appear well separated, and the azimuthal velocity time series at the CO site confirms this by showing a circulation pattern inconsistent with either IHE. Therefore, the text should clarify how the authors define when a site passes "near the location of a previously detected eddy". The solid line along the CO trajectory and the dashed lines in the Hovmöller diagram (Figure 9b) are misleading based on the data shown. Removing or redefining these visual cues would help reduce confusion.
**We have rewritten the sentence to avoid implying that the CO site passed close to either eddy. As suggested, we now explicitly state that the sADCP velocity field shows no signature associated with either IHE during this period, indicating that CO did not intersect their circulation: "The sADCP from Polarstern (Figure 10b, right panel) shows no velocity signature associated with either eddy during the period marked by the green dashed line. Therefore, the two eddies appear dynamically independent, and the CO site did not intersect the azimuthal circulation of either feature during its drift".**

- L278-283: The authors conclude that E5 and E6 correspond to the same eddy sampled about one week apart. This interpretation should be developed further, either in the Discussion or as a perspective for future work. Questions naturally arise: To what extent did the eddy evolve during this period? What mechanisms could have influenced or forced such changes (e.g., erosion, intensification, or mixing)? In L280–282, there is also an apparent contradiction. The text first suggests that differences between E5 and E6 arise because different water masses of the eddy were sampled, yet immediately afterward states that Figure 10 shows both structures share the same water-mass properties. Please clarify this point or rephrase to resolve the inconsistency. Finally, note that besides L3, the CO site appears to have crossed these structures—possibly earlier than L3—yet the azimuthal velocity Hovmöller does not clearly display the eddy signal. This aspect could be acknowledged or briefly discussed, as it adds context to the spatial and temporal sampling coverage.

**We now explicitly state that intrahalocline eddies are expected to evolve slowly, with previous observations suggesting lifetimes of at least 21 months (e.g., Zhao et al., 2014). This supports the interpretation that the eddy did not undergo a substantial change of core-centre properties between the two sampling events. We now clarify that the thermohaline differences between E5 and E6 are minimal (order 0.01 in temperature and salinity), and are best explained by the fact that the two crossings did not transect exactly the same portion of the eddy, rather than by physical evolution. We now mention that the CO crossed the area where the eddy was found, but no eddy signal was found, most likely due to the earlier measurement: "Site L3 recorded two eddies within a week, with centres separated by 6 km. This suggests that both detections correspond to the same eddy, which would have a translation speed of approximately 0.01 $ms^{-1}$ during that period. The thermohaline and kinematic properties support this interpretation (Table 2): the differences between E5 and E6 are minimal (order 0.01 in $\Theta$ and $S_A$) and are consistent with the instruments not sampling the same cross-section of the eddy, which also explains the moderate difference in estimated radius (7.3-5.2 km). The $\Theta$-$S_A$ structure (Figure 12c) confirms that both features share nearly identical core water masses, indicating no appreciable modification of the eddy over the one-week interval. This is expected, as intrahalocline eddies can persist for extended periods. Zhao et al. (2014) report a lifespan of at least 21 months for Eurasian Basin IHEs, and therefore their thermohaline structure is not expected to change substantially on weekly timescales. Although the CO site passed near the region where the eddy was located, no clear azimuthal-velocity anomaly was detected (Figure 10b, purple line), likely due to an earlier partial crossing or an incomplete intersection with the eddy core".**

- L290: The text states that the observed IHEs are "saltier and warmer" than those described by Zhao et al. (2014). However, Figure 10b does not

clearly show them as warmer—rather, they appear to deviate from the T–S relationship established in Zhao et al. (2014). Please clarify this interpretation: are the eddies indeed warmer in absolute terms, or do they simply depart from the previously reported temperature–salinity trend?

**We now clarify that the eight eddies identified in this study correspond to shallow Eurasian Basin intrahalocline eddies, which contain saltier waters that are less close to the freezing point compared to those reported by Zhao et al. (2014). As a result, our eddies depart from the temperature–salinity relationship described in that study, rather than being uniformly warmer in an absolute sense. This distinction is now clearly stated in the revised manuscript and highlighted in Figure 10b.**

- L290-292: When describing the hydrographic structures, it would be helpful to guide the reader through the specific region of the T–S diagram being discussed—indicating the approximate temperature and salinity ranges where they are located. Additionally, please expand qualitatively on the meaning of the term "wedge shape."

**We clarified the description by defining more explicitly what we mean by a "wedge shape" in the $\Theta$-$S_A$ diagram. We explain that this structure corresponds to a concave curve in which temperature decreases toward a local minimum at the eddy core-centre and then increases again while salinity continues to rise. We now also specify the approximate temperature and salinity ranges where this pattern appears for the eddies discussed: "The $\Theta$-$S_A$ diagram in Figure 12 shows three different characteristic shapes: (i) fluctuant temperature with a smooth "wedge" shape in E8 and E9 (Figure 12a) located approximately within the range of $-1.74 \pm 0.03°C$ and $33.62 \pm 0.1$ $gkg^{-1}$, (ii) a smoother curve in October (Figure 12b) and (iii) a prominent "wedge" shape in November (Figure 12c) around $-1.81 \pm 2°C$ and $34 \pm 0.3$ $gkg^{-1}$. By "wedge shape," we refer to a T–S structure in which temperature decreases toward a local minimum at the eddy core-centre, reaches a local minimum, and then increases again as salinity continues to rise, forming a characteristic concave shape in the diagram".**

- L321-322: The statement "given the lack of strong horizontal velocity shear... barotropic instability is unlikely to be a dominant mechanism in this region" is not explicitly supported by the presented observations nor previous literatrure. No quantitative evidence of horizontal velocity shear is shown, and the match between eddy scale and alone is insufficient to exclude barotropic processes. If this conclusion cannot be demonstrated directly, I suggest either removing the sentence or reframing it as a hypothesis or qualitative inference to be tested in future studies.

**We have removed the sentence suggesting that barotropic instability is unlikely, as we agree that the available observations do not allow us to quantitatively assess horizontal velocity shear or conclusively rule**

**out barotropic processes. The discussion now focuses on mechanisms for which supporting evidence is available.**

- Figure 10: I recommend dividing this figure into two separate figures—one containing panels (a) and (b), and another with panels (c–e). This separation would improve readability and allow each set of results (hydrographic vs. dynamical) to be interpreted more clearly.
**Done.**

[Figure]

Figure 11: $\Theta$-$S_A$ diagrams. Density contours and freezing temperature (at surface pressure) are shown in grey dashed lines and in blue dashed lines, respectively. Mean profiles of the surrounding water are shown in grey, and the colours represent each eddy up to 90 meters depth with the core-centre properties shown by the larger markers. b) shows the core-centre $\Theta$-$S_A$ values, and in pink the core-centre values of Eurasian eddies from Zhao et al. (2014).

[Figure]

Figure 12: Θ-$S_A$ diagrams. Density contours and freezing temperature (at surface pressure) are shown in grey dashed lines and in blue dashed lines, respectively. Mean profiles of the surrounding water are shown in grey, and the colours represent each eddy up to 90 meters depth with the core-centre properties shown by the larger markers. Groups of eddies with similar Θ-$S_A$ curves indicating different generation processes: a) refreezing and convection, b) advective-convective and c) convective cold halocline.

- L325: Specify that eight IHEs were identified in total, one of which was characterised twice, to make the count explicit and consistent with the preceding description.
**Done.**

- L338: The proposed observational resolution of 5 km seems inconsistent with the reported eddy radii (6 km app). If the goal is to resolve the smaller IHEs, a finer spatial resolution would be required.
**We agree with this observation. We now clarify that resolving intra-halocline eddies of typical radius ≈ 6 km requires a finer observational spacing than previously stated. The manuscript has been revised to recommend higher-resolution (2–3 km) spatially distributed autonomous measurements, which would allow both adequate resolution of the eddy structure and repeated sampling of individual features.**